



Climate
of the Past

# Late Holocene (0–6 ka) sea-level changes in the Makassar Strait, Indonesia

**Maren Bender**[1], **Thomas Mann**[2], **Paolo Stocchi**[3], **Dominik Kneer**[4], **Tilo Schöne**[5], **Julia Illigner**[5], **Jamaluddin Jompa**[6], **and Alessio Rovere**[1]

[1]MARUM – Center for Marine Environmental Sciences, University Bremen, Leobener Straße 8, 28359 Bremen, Germany
[2]ZMT – Leibniz Centre for Tropical Marine Research, Fahrenheitsstraße 6, 28359 Bremen, Germany
[3]NIOZ – Royal Netherlands Institute for Sea Research, 17907 SZ 't Horntje, Texel, the Netherlands
[4]Alfred Wegener Institute, Helmholtz Centre for Polar and Marine Research, Hafenstrasse 43, 25992 List/Sylt, Germany
[5]Helmholtz-Zentrum Potsdam – Deutsches GeoForschungsZentrum (GFZ), Telegrafenberg 14473 Potsdam, Germany
[6]Graduate School, Hasanuddin University, Makassar, 90245, Indonesia

**Correspondence:** Maren Bender (mbender@marum.de)

**Abstract.** The Spermonde Archipelago, off the coast of southwest Sulawesi, consists of more than 100 small islands and hundreds of shallow-water reef areas. Most of the islands are bordered by coral reefs that grew in the past in response to paleo relative sea-level changes. Remnants of these reefs are preserved today in the form of fossil microatolls. In this study, we report the elevation, age, and paleo relative sea-level estimates derived from fossil microatolls surveyed in five islands of the Spermonde Archipelago. We describe 24 new sea-level index points, and we compare our dataset with both previously published proxies and with relative sea-level predictions from a set of 54 glacial isostatic adjustment (GIA) models, using different assumptions on both ice melting histories and mantle structure and viscosity. We use our new data and models to discuss Late Holocene (0–6 ka) relative sea-level changes in our study area and their implications in terms of modern relative sea-level estimates in the broader South and Southeast Asia region.

## 1 Introduction

After the Last Glacial Maximum, sea level rose as a result of increasing temperatures and ice loss in polar regions. Rates of sea-level rise due to ice melting and thermal expansion (i.e., eustatic) progressively decreased between 8 to 2.5 ka (Lambeck et al., 2014), remaining constant thereafter (until the post-industrial sea-level rise). In areas far from polar regions (i.e., far-field; Khan et al., 2015) the rapid eustatic sea-level rise after the Last Glacial Maximum was followed by a local (i.e., relative) sea-level highstand between ∼ 6 and ∼ 3 ka, and a subsequent sea-level fall towards present-day sea level. It has long been shown that the higher-than-present relative sea level (RSL) in the Middle Holocene (e.g., Grossman et al., 1998; Mann et al., 2016) is not eustatic in origin but was caused by the combined effects of glacial isostatic adjustment (GIA) (Milne and Mitrovica, 2008), which includes ocean siphoning (Milne and Mitrovica, 2008; Mitrovica and Milne, 2002; Mitrovica and Peltier, 1991) and redistribution of water masses due to changes in gravitational attraction and Earth rotation following ice mass loss (Kopp et al., 2015).

Due to the spatiotemporal variability of the processes causing it, the Late Holocene highstand differs regionally in both time and elevation. The occurrence and elevation of RSL indicators deposited during the highstand are dependent not only on the processes mentioned above but also on the magnitude of Holocene land-level changes due to geological processes, such as subsidence resulting from sediment compaction or tectonics (e.g., Tjia et al., 1972; Zachariasen, 1998). Combining the use of precisely measured and dated RSL indicators with GIA models in areas where the highstand occurs, it is possible to improve our knowledge on long-term rates of land-level changes, which need to be con-

sidered in conjunction with local patterns and rates of current eustatic sea-level rise (e.g., Dangendorf et al., 2017) to gauge the sensitivity of different areas to future coastal inundation.

In this study, we present new Late Holocene sea-level data and GIA models from the Spermonde Archipelago (Central Indonesia, SW Sulawesi). In this region, a recent review (Mann et al., 2019a, b) indicated discrepancies between the RSL data reported by different studies. To reconstruct the local paleo RSL we surveyed microatolls, i.e., particular coral morphologies forming in close connection with sea-level datums (e.g., Scoffin and Stoddart, 1978; Woodroffe et al., 2012, 2014). For reconstructing paleo RSL, we first studied living coral microatolls to calculate the range of depth where corals live at different islands. We then applied the results of the living microatoll (LMA) survey to fossil ones that we surveyed and dated using radiocarbon.

In total, we surveyed 24 fossil microatolls (FMAs), with ages clustered around ∼ 155 and ∼ 5000 years before present (BP). We present this new dataset in conjunction with data provided by previous studies in the same region (Mann et al., 2016; Tjia et al., 1972; De Klerk, 1982) and new GIA models with varying ice histories and mantle properties. We use our data and models to discuss possible local subsidence mechanisms at the only heavily populated island (Barrang Lompo) among those we investigated, vertical land movements in the broader Spermonde Archipelago, and implications of the different ice and earth models for modern relative sea-level change estimates.

## 2   Regional setting

The Spermonde Archipelago, located between $4°00'$ to $6°00'$ S and $119°00'$ to $119°30'$ E, hosts several low-lying islands, with average elevations of 2 to 3 m above mean sea level (Janßen et al., 2017; Kench and Mann, 2017). All these islands consist of table, platform, and patch reefs crowned by coral cays (Sawall et al., 2011) and some are densely populated (Schwerdtner Máñez et al., 2012). Their low elevation above mean sea level (MSL) and the fact that they are composed mostly of calcareous sediments makes them vulnerable to sea-level rise, inundation by waves, and deficits in sediment supply (Kench and Mann, 2017). In the Spermonde Archipelago, the tidal cycle is mixed semi-diurnal with a maximum tidal range of 1.5 m (data from Badan Informasi Geospasial, Indonesia).

In this study, we focused on five islands in the Spermonde Archipelago. Here, we surveyed fossil microatolls that are complementary to those previously surveyed at two other islands in the same archipelago, reported in Mann et al. (2016) (Fig. 1a, b). Panambungan (RSL data in Mann et al., 2016) (Fig. 1g) is a small and uninhabited island, located 18 km northwest of Makassar City. Barrang Lompo (RSL data in Mann et al., 2016) (Fig. 1i) is located 11.2 km northwest of Makassar and 11 km southwest of Panambungan, and is

densely populated. Bone Batang (Fig. 1h) is a narrow, uninhabited sandbank located south of the island of Panambungan and north of the island of Barrang Lompo. South of Barrang Lompo and 13 km southwest from the city of Makassar, we surveyed Kodingareng Keke (Fig. 1c), another uninhabited island. The island of Sanrobengi (Fig. 1d) lies 25 km south of Kodingareng Keke, and is a small, sparsely inhabited (less than 15 houses) reef island located close to the mainland of southern Sulawesi at the coast of Galesong, 21 km south of Makassar city. Sanrobengi is located south of the previous islands, which are close to each other off the coast of Makassar, towards the center of the archipelago. The fourth and fifth study islands are located northwest of Makassar, bordering the edge of the Spermonde Archipelago. These two outer islands are Suranti (Fig. 1f) and Tambakulu (Fig. 1e) and both are uninhabited and located 58 km (Suranti) and 56 km (Tambakulu) from the City of Makassar. Another island already reported and studied by Mann et al. (2016) (Sanane) is included in this study only for the analysis of living microatolls, as fossil microatolls were not found on this island. Its location is 2.7 km northwest of Panambungan, and it is densely populated.

## 3   Methods

### 3.1   Coral microatolls

In most tropical areas, Holocene RSL changes can be reconstructed using several types of RSL indicators (Khan et al., 2015), among which are fossil coral microatolls (e.g., Scoffin and Stoddart, 1978; Woodroffe et al., 2012; Woodroffe and Webster, 2014). Fossil microatolls are particular growth forms adopted by massive corals (e.g., *Porites*) when they reach the upper bounds of their living range, close to sea level. Coral colonies generally grow upwards until they reach the lower part of the tidal range. At this point, they keep growing horizontally forming "atoll-like" structures (Fig. 1 in Scoffin and Stoddart, 1978, and Fig. 8.1 in Meltzner and Woodroffe, 2015) that can widen up to several meters.

In the most standard definition, microatolls typically form at mean lower low water (MLLW), but their living range can span from mean low water (MLW) down to the lowest astronomical tide (LAT) (Mann et al., 2019a). If sea level falls below LAT, the coral polyps desiccate and die, retaining their carbonate calcium skeleton and their morphology (Meltzner and Woodroffe, 2015). Since they can survive within a narrow range related to tidal datums, fossil microatolls are often considered as an excellent RSL indicator (when found in good preservation state) as they constrain paleo RSL within a narrow range (Meltzner and Woodroffe, 2015).

While the relationship of coral microatolls with the tidal datums described above is often maintained, several authors (e.g., Mann et al., 2016; Smithers and Woodroffe, 2001; Woodroffe et al., 2012) have pointed out that deviations from microatoll living range and tidal datums may occur due to

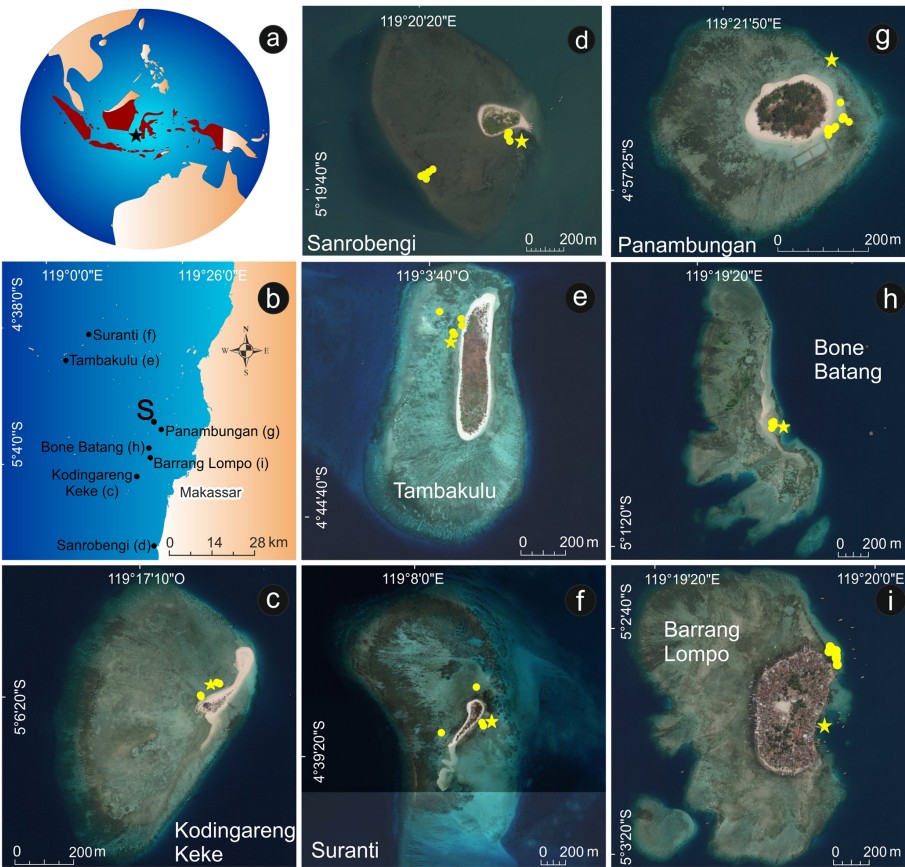

**Figure 1.** Overview map of the islands investigated in this study and the two islands studied by Mann et al. (2016) (Panambungan and Barrang Lompo). The star in **(a)** indicates the location of the Spermonde Archipelago, off the coast of southwestern Sulawesi; **(b)** indicates the position of each island, and the dot labeled "S" indicates the position of Sanane, where only living microatolls were surveyed. Insets **(c–i)** show each island. The yellow dots in these panels indicate the location of sampled fossil microatolls, while the yellow asterisks indicate the position of the tide pressure sensor. Imagery sources for panels **(a, b)**: Global Self-consistent Hierarchical High-resolution Shorelines from Wessel and Smith (1996); for **(c–i)**: Esri, DigitalGlobe, GeoEye, Earthstar Geographics, CNES/Airbus DS, USDA, USGS, AeroGRID, IGN, and the GIS User Community. The background maps in Fig. 1 were created using ArcGIS® software by Esri. ArcGIS® and ArcMap™ are the intellectual property of Esri and are used herein under license. Copyright© Esri. All rights reserved. For more information about Esri® software, please visit https://www.esri.com/en-us/home (last access: 11 May 2020).

site-dependent characteristics, such as wave intensity and broader reef morphology (Meltzner and Woodroffe, 2015). It is also worth highlighting that a tide gauge with a long enough time series might not be available at remote locations where microatolls are often found. Therefore, it is often considered more practical and more accurate to reconstruct paleo RSL at the time of microatoll life starting from the height of living coral microatolls (HLC; see Meltzner and Woodroffe, 2015, for details). Under the assumption that tide, wave, and reef morphology did not change significantly in time, this allows determining the paleo RSL associated with fossil microatolls that were living in the same geographical setting as modern ones (i.e., the same island or group of islands). For this reason, in this study, we surveyed both fossil and living microatolls elevations, and we determined the indicative

meaning (i.e., the correlation with sea level) of the fossil microatolls from the HLC rather than tidal datums.

As fossil microatolls are composed of calcium carbonate, they can be assigned an age, either with $^{14}C$ (Woodroffe et al., 2012) or U-series dating (e.g., Azmy et al., 2010). Recent studies have shown that the accurate measurement, dating, and standardized interpretation of coral microatolls have the further potential to detail patterns and cyclicities related to short-term (e.g., decadal to centennial) sea-level fluctuations (Meltzner et al., 2017; Smithers and Woodroffe, 2001; Kench et al., 2019).

### 3.2 Elevation measurements

FMA and LMA heights were surveyed on Sanrobengi, Kodingareng Keke, Bone Batang, Suranti and Tambakulu

(Fig. 1c–i) with an automatic level. FMA and LMA heights were always surveyed on the top microatoll surface. Elevations were initially referenced to locally deployed water level sensors (Seametrics PT2X) acting as temporary benchmarks. Locations of water level loggers are shown in Fig. 1c–i (stars), and logged water levels are reported in the PANGAEA dataset associated with this paper. The sensors were fixed to either jetties or living corals close to the survey sites and logged the tide levels at 30 s intervals. Tidal level differences between the sensors on the study islands were referenced to the tidal height of the water level sensor on Panambungan, for which we have the longest tide record of 8 d and 18 h. The Panambungan tidal readings were compared to readings at the national tide gauge at Makassar harbor (1 January 2011–19 December 2019; data courtesy of Badan Informasi Geospasial, Indonesia) to establish the reference of our sample sites to MSL. As a result of annual sea-level variability, the mean tidal level at Makassar during our surveys was slightly above (+0.014 m) the long-term MSL (1 January 2011 to 19 December 2019). Our elevation measurements were corrected accordingly.

FMA and LMA measurement errors were propagated using the root mean square of the sum of squares of the following values (see the PANGAEA dataset associated with this paper for calculations and details):

- automatic level survey error = 0.02 m, as in Mann et al. (2016). If the automatic level had to be moved due to excessive distance from the benchmark to the measured point, this error is added twice.

- error referencing island logger to Panambungan MSL. This error has been calculated comparing water levels measured at each island against those measured at Panambungan and varies from 0.01 to 0.07 m (see the PANGAEA dataset associated with this paper for details).

- error referencing Panambungan to Makassar MSL = 0.04 m, as in Mann et al. (2016).

- error in calculating Makassar MSL from a limited time (8.9 years, 1 January 2011 to 19 December 2019) and not for an entire tidal cycle (18.6 years). We estimated this error to be 0.05 m.

### 3.3 Paleo RSL calculation

After relating all microatoll elevations to MSL, we used FMA and LMA elevation measurements to calculate paleo RSL. We then applied the concept of indicative meaning (see Shennan, 1986, for definition and applications) to coral microatolls. The indicative meaning allows quantifying the relationship between the RSL indicator and the associated paleo sea level. To reconstruct paleo RSL from measured data we use the following formula:

$$RSL = E - HLC + Er,$$

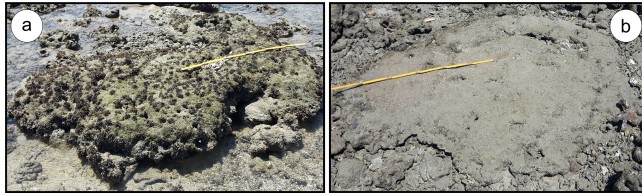

**Figure 2.** Examples of **(a)** non-eroded and **(b)** eroded fossil microatoll at Sanrobengi.

where $E$ is the surveyed elevation of the fossil microatoll; HLC is the average height of living coral microatolls, and Er is the estimated portion that was eroded from the upper fossil microatoll surface.

To calculate RSL, we measured HLC at each island individually or at the closest neighboring island where living microatolls could be found.

Concerning HLC, we surveyed living microatolls on Tambakulu (samples $n = 51$) and Sanrobengi ($n = 24$). On Suranti, Kodingareng Keke, and Bone Batang, living microatolls were either restricted in number and with a partly reworked appearance or completely absent. Therefore, to calculate RSL at these islands, we used HLC elevations from Tambakulu ($n = 51$) for Suranti, from Panambungan (from Mann et al., 2016; $n = 20$) for Bone Batang, and from Barrang Lompo (from Mann et al., 2016; $n = 23$) for Kodingareng Keke.

The Er value was included in our calculation only in the presence of visibly eroded microatolls (see Table 2 for details, comparison with non-eroded microatolls in Fig. 2a, b) to account for the lowering of the top microatoll surface due to erosion. In Fig. 3a and b, these microatolls are indicated with surrounding light gray shading. Measurements on modern microatolls at Barrang Lompo, Panambungan, and Sanane (Fig. 4a) by Mann et al. (2016) showed that the average thickness of living microatolls in the Spermonde Archipelago is $0.48 \pm 0.19$ m. Thus, to reconstruct the original fossil microatoll elevation for eroded FMAs, we added the missing centimeters to each eroded FMA thickness to reach 0.48 m. We remark that this approach does not take into account the fact that modern microatolls may be thicker than fossil ones because of the current rapidly rising sea level (which is forcing them to catch up, growing faster upwards). In contrast, under Late Holocene falling or stable sea-level changes, they were presumably getting wider but not thicker. Hence, in our calculations, the added Er might be overestimated. In the absence of better constraints, we maintain this approach.

Final paleo RSL uncertainties were calculated using the root mean square of the sum of squares of the following values (see the PANGAEA dataset associated with this paper for calculations and details):

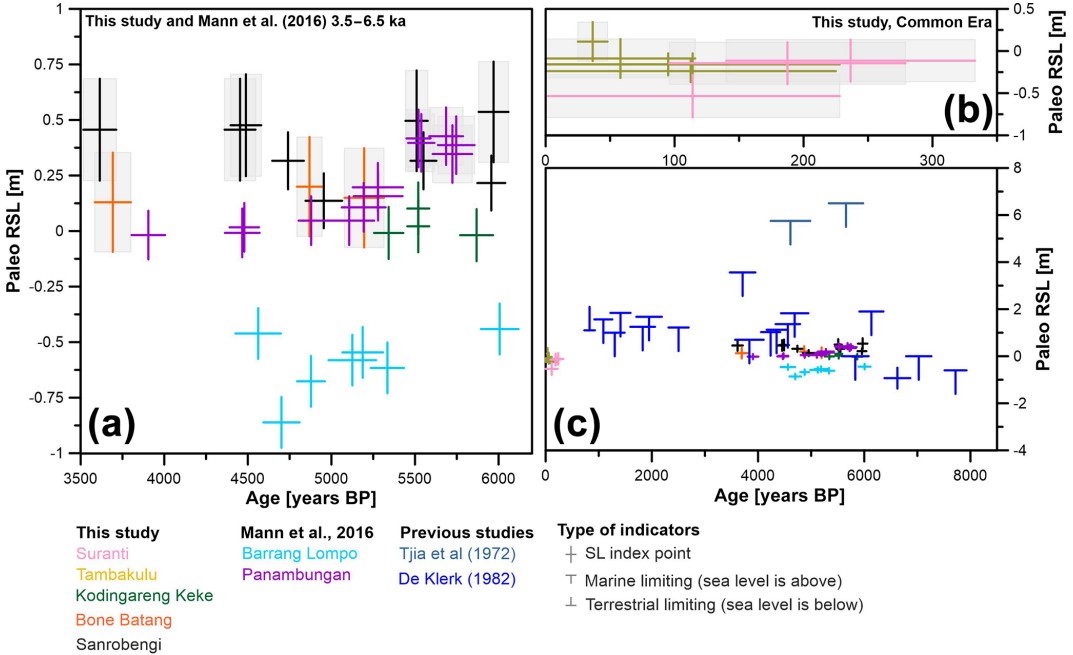

**Figure 3.** Representation of data reported in Tables 2 and 3. **(a)** RSL index points dating ∼ 6.5 to ∼ 3.5 ka and **(b)** Common Era microatolls surveyed in this study. Gray bands in **(a, b)** represent the microatolls that were recognized as eroded in the field and to which the erosion correction explained in the text has been applied. Panel **(c)** shows the newly surveyed data in the context of previous studies.

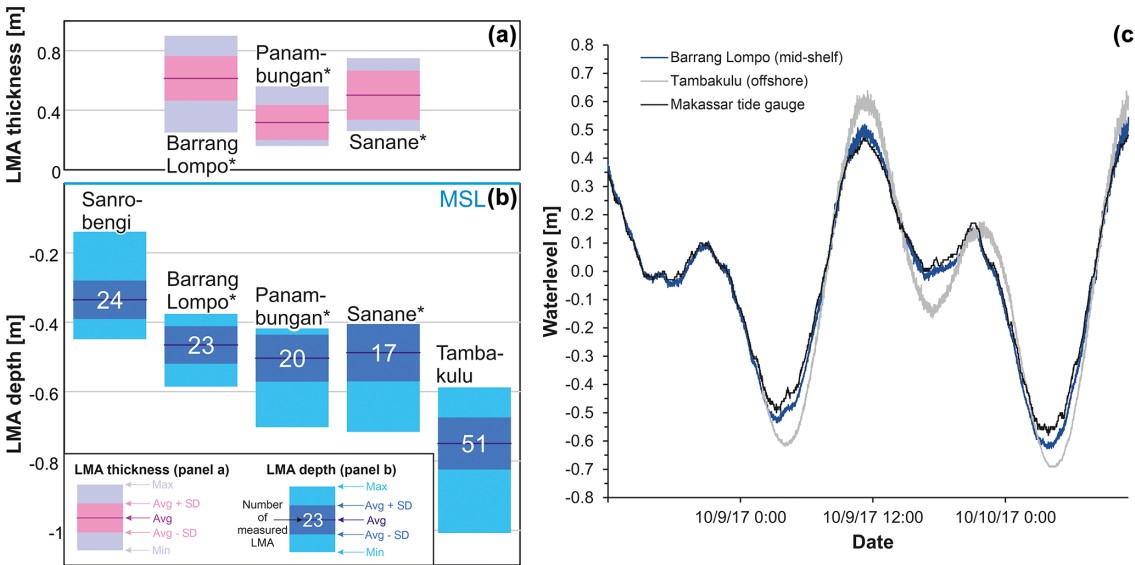

**Figure 4. (a)** Thickness of living microatolls (LMAs) measured by Mann et al. (2016) in the Spermonde Archipelago. The average of the three islands reported is $0.48 \pm 0.19$ m. **(b)** Measured depth of LMA in this study (Sanrobengi and Tambakulu islands) and in Mann et al. (2016). The asterisk in **(a, b)** indicates the islands surveyed by Mann et al. (2016). In **(a, b)** the islands are ordered from that closest to the shore on the left side to those further away from the shore on the right side. **(c)** Comparison between water levels measured at Barrang Lompo (located on the mid-shelf), Tambakulu (located offshore towards the edge of the shelf) and data recorded by the national tide gauge at Makassar harbor. Note that, in **(a, b)**, "zero" refers to mean sea level, while in **(b)** "zero" refers to the average water level over the measurement period (here 8–10 October 2017).

– elevation errors of both FMA and LMA, calculated as described above

– half of the indicative range, represented by the standard deviation of the measured heights of living corals

– uncertainty in estimating erosion $= 0.19$ m, derived from Mann et al. (2016) and discussed above.

### 3.4   Sampling and dating

The highest point of each FMA was sampled by hammer and chisel or with a hand drill. Subsamples from all samples
taken in the field were analyzed via XRD at the Central Laboratory for Crystallography and Applied Material Sciences (ZEKAM), University of Bremen, Germany, to detect possible diagenetic alterations of the aragonite coral skeleton.

After the XRD screening, we performed one radiocarbon
dating per sampled microatoll. AMS radiocarbon dating and age calibration to calendar years before present (years BP) were done at Beta Analytic Laboratory. We used the Marine 13 calibration curve (Reimer et al., 2013) and a $\delta R$ value (the reservoir age of the ocean) of $0 \pm 0$ as recommended for
Indonesia in Southon et al. (2002). To compare the new ages to the results from Mann et al. (2016), we recalculated their ages with the same $\delta R$ value.

The reason behind choosing a different $\delta R$ value than Mann et al. (2016) resides in the fact that the value they
adopted ($\delta R = 89 \pm 70$) was measured in southern Borneo (Southon et al., 2002) more than 900 km away from our study site. Their choice was based on the fact that there is no $\delta R$ value available between Sulawesi and southern Borneo that can be used for a radiocarbon age reservoir correction. Due
to the long distance between Borneo and our study area and the presence of the Indonesian Throughflow between these two regions (Fieux et al., 1996), here we propose that there is no basis to assume a similar $\delta R$ value between southern Borneo and the Spermonde Archipelago. Therefore we fol-
low the recommendation of Southon et al. (2002) to use a zero $\delta R$, reported to be derived from unpublished data for the Makassar Strait.

All our samples were registered in the SESAR, the System for Earth Sample Registration, and assigned an International
Geo-Sample Number (IGSN).

### 3.5   Glacial isostatic adjustment

To compare observations with RSL caused by isostatic adjustment since the Last Glacial Maximum, we calculated RSL as predicted by geophysical models of GIA. These
45 are based on the solution of the sea-level equation (Clark and Farrell, 1976; Spada and Stocchi, 2007). We calculate GIA predictions using a suite of combinations of ice sheets and solid Earth models. The latter are self-gravitating, rotating, radially stratified, deformable, and characterized by
50 a Maxwell viscoelastic rheology. We discretize the Earth's

**Table 1.** Upper and lower mantle viscosities for the different Earth models.

| Model name | Upper mantle (Pa s $\times 10^{21}$) | Lower mantle (Pa s $\times 10^{21}$) |
|---|---|---|
| VM1 | 0.25 | 2.5 |
| VM2 | 0.25 | 5.0 |
| VM3 | 0.25 | 10 |
| VM4 | 0.5 | 2.5 |
| VM5 | 0.5 | 5 |
| VM6 | 0.5 | 10 |

mantle into two layers: upper and lower mantle (respectively, UM and LM). Each mantle viscosity profile is combined with a perfectly elastic lithosphere whose thickness is set to 60, 90, or 120 km. We use six mantle viscosities for each lithospheric thickness, as shown in Table 1. We combine the Earth
models with three different models: ICE5g, ICE6g (Peltier et al., 2015; Peltier, 2009), and ANICE (de Boer et al., 2015, 2017). In total, we ran 54 different ice–Earth model combinations (three ice sheet models × three lithospheric thicknesses × six mantle viscosity profiles).

## 4   Results

### 4.1   Living and fossil microatolls

Our dataset consists of a total of 25 FMAs surveyed in five islands of the Spermonde Archipelago (Table 2; see also the PANGAEA dataset associated with this paper). Sixteen mi-
65 croatolls yield ages (calendar years) ranging from 5970 to 3615 years BP (Fig. 3a), while nine yield ages varying from 237 to 37 years BP (Fig. 3b). These are added to the 20 fossil microatolls and one modern microatoll from Barrang Lompo and Panambungan previously reported by Mann et al. (2016)
(Fig. 3a, c; see also the PANGAEA dataset associated with this paper) and the data from De Klerk (1982) and Tjia et al. (1972) (Fig. 3c and Table 4, see also the PANGAEA dataset associated with this paper). The microatoll PS_FMA 4 showed evidence of reworking; i.e., it was not fixed to the
sea bottom, and thus it was subsequently rejected. Therefore, it is not shown in the results or discussed further. Among the 44 microatolls surveyed and dated in this study ($n = 24$) and Mann et al. (2016) ($n = 20$), 18 were eroded, and the erosion correction has been applied as reported in Sect. 3.3 (gray
bands in Fig. 3a). The fact that these corrected data seem to plot consistently above the non-eroded microatolls might be indicative of the fact that our erosion correction may be overestimated. In the absence of more precise data on the original thickness of fossil microatolls, we retain these indicators in
our analyses.

Concerning LMAs, our surveys included 51 individuals measured at the island of Tambakulu and 24 living microatolls measured at Sanrobengi (Fig. 4b). The living microa-

**Table 2.** Fossil microatolls surveyed and dated at Suranti (PS_FMA1–3), Tambakulu (PT_FMA5–9), Bone Batang (BB_FMA11–13), Kodingareng Keke (KK_FMA14–17), and Sanrobengi (SB_FMA18–26). All ages are recalculated with the $\delta R$ value of $0 \pm 0$ (Southon et al., 2002). The elevation–age plot of these data is shown in Fig. 3a and b.

| IGSN | Lab code | Sample name | Island name | $^{14}$C age | $\pm^{14}$C error | Mean age (cal yr BP) | $\pm$ Error (yr) | Elevation (m) with respect to m.s.l. | HLC (m) | RSL (m) | $\pm$ Vertical error (m) | $+$ Erosion error ($\sigma$Er) (m) |
|---|---|---|---|---|---|---|---|---|---|---|---|---|
| IEMBMPSFMA1 | Beta – 487554 | PS_FMA1 | Suranti | 490 | 30 | 114 | 114 | −1.48 | −0.75 | −0.53 | 0.25 | 0.2 |
| IEMBMPSFMA2 | Beta – 508373 | PS_FMA2 | Suranti | 560 | 30 | 187.5 | 91.5 | −1.22 | −0.75 | −0.14 | 0.25 | 0.33 |
| IEMBMPSFMA3 | Beta – 487555 | PS_FMA3 | Suranti | 620 | 0 | 236.5 | 96.5 | −1.19 | −0.75 | −0.11 | 0.25 | 0.33 |
| IEMBMPTFMA5 | Beta – 487558 | PT_FMA5 | Tambakulu | 460 | 30 | 95 | 95 | −0.91 | −0.75 | −0.16 | 0.13 | 0 |
| IEMBMPTFMA6 | Beta – 508375 | PT_FMA6 | Tambakulu | 490 | 30 | 114 | 114 | −0.91 | −0.75 | −0.16 | 0.13 | 0 |
| IEMBMPTFMA7 | Beta – 508376 | PT_FMA7 | Tambakulu | 470 | 30 | 112.5 | 112.5 | −0.99 | −0.75 | −0.24 | 0.13 | 0 |
| IEMBMPTFMA8 | Beta – 487559 | PT_FMA8 | Tambakulu | 106.55 | 0.4 pMC | 36.5 | 11.5 | −0.84 | −0.75 | 0.11 | 0.23 | 0.2 |
| IEMBMPTFMA9 | Beta – 508377 | PT_FMA9 | Tambakulu | 420 | 30 | 58 | 58 | −0.97 | −0.75 | −0.09 | 0.23 | 0.13 |
| IEMBMBBFMA11 | Beta – 487545 | BB_FMA11 | Bone Batang | 4630 | 30 | 4869 | 75 | −0.58 | −0.50 | 0.20 | 0.22 | 0.28 |
| IEMBMBBFMA12 | Beta – 487546 | BB_FMA12 | Bone Batang | 4910 | 30 | 5196 | 118 | −0.65 | −0.50 | 0.15 | 0.22 | 0.3 |
| IEMBMBBFMA13 | Beta – 508378 | BB_FMA13 | Bone Batang | 3750 | 30 | 3692.5 | 107.5 | −0.67 | −0.50 | 0.13 | 0.22 | 0.3 |
| IEMBMKKFMA14 | Beta – 487556 | KK_FMA14 | Kodingareng Keke | 4970 | 30 | 5342.5 | 87.5 | −0.47 | −0.47 | −0.01 | 0.12 | 0 |
| IEMBMKKFMA15 | Beta – 508379 | KK_FMA15 | Kodingareng Keke | 5500 | 30 | 5868.5 | 98.5 | −0.48 | −0.47 | −0.02 | 0.12 | 0 |
| IEMBMKKFMA16 | Beta – 487557 | KK_FMA16 | Kodingareng Keke | 5160 | 30 | 5519.5 | 65.5 | −0.36 | −0.47 | 0.10 | 0.12 | 0 |
| IEMBMKKFMA17 | Beta – 508380 | KK_FMA17 | Kodingareng Keke | 5160 | 30 | 5519.5 | 65.5 | −0.44 | −0.47 | 0.02 | 0.12 | 0 |
| IEMBMSBFMA18 | Beta – 487547 | SB_FMA18 | Sanrobengi | 4730 | 30 | 4954.5 | 109.5 | −0.20 | −0.34 | 0.14 | 0.12 | 0 |
| IEMBMSBFMA19 | Beta – 508371 | SB_FMA19 | Sanrobengi | 5560 | 30 | 5956.5 | 83.5 | −0.12 | −0.34 | 0.22 | 0.12 | 0 |
| IEMBMSBFMA20 | Beta – 487548 | SB_FMA20 | Sanrobengi | 5140 | 30 | 5509.5 | 66.5 | −0.17 | −0.34 | 0.50 | 0.23 | 0.33 |
| IEMBMSBFMA21 | Beta – 487549 | SB_FMA21 | Sanrobengi | 5570 | 30 | 5970 | 89 | −0.13 | −0.34 | 0.54 | 0.23 | 0.33 |
| IEMBMSBFMA22 | Beta – 487550 | SB_FMA22 | Sanrobengi | 5200 | 30 | 5550.5 | 77.5 | −0.02 | −0.34 | 0.32 | 0.13 | 0 |
| IEMBMSBFMA23 | Beta – 487551 | SB_FMA23 | Sanrobengi | 4550 | 30 | 4740.5 | 94.5 | −0.02 | −0.34 | 0.32 | 0.13 | 0 |
| IEMBMSBFMA24 | Beta – 487552 | SB_FMA24 | Sanrobengi | 4350 | 30 | 4488.5 | 91.5 | −0.01 | −0.34 | 0.48 | 0.23 | 0.15 |
| IEMBMSBFMA25 | Beta – 487553 | SB_FMA25 | Sanrobengi | 4320 | 30 | 4453.5 | 92.5 | −0.03 | −0.34 | 0.46 | 0.23 | 0.15 |
| IEMBMSBFMA26 | Beta – 508372 | SB_FMA26 | Sanrobengi | 3700 | 30 | 3614.5 | 98.5 | −0.03 | −0.34 | 0.46 | 0.23 | 0.15 |

tolls in this survey complement those measured by Mann et al. (2016) at Panambungan ($n = 20$), Barrang Lompo ($n = 23$) and Sanane islands ($n = 17$).

To reference the measured elevations of both LMA and FMA to MSL as described in Sect. 3.3, we measured water levels at Barrang Lompo, Panambungan, Suranti, Tambakulu, Kodingareng Keke, Bone Batang, and Sanrobengi for a total of 688 h, over the period 6 to 15 October 2017 (see water levels in the PANGAEA dataset associated with this paper). An example of measured water levels is shown in Fig. 4b.

As far as XRD analyses are concerned (see the PANGAEA dataset associated with this paper for details), 17 of 24 samples show an average value of aragonite at $98.7 \pm 1.1\%$. Among the other samples, one (SB_FMA26) contains 7 % calcite, which might affect its age. Other potential sources of secondary carbon might be present in PT_FMA9 and BB_FMA13, where Kutnohorite was detected ($CaMn^{2+}(CO_3)^2$, respectively 3 % and 6 %). All the remaining samples show relatively low aragonitic content, but the other minerals contained in them do not contain carbon that could potentially affect the ages reported in this study.

The fossil microatolls of Suranti show age ranges from $237 \pm 97$ to $114 \pm 114$ years BP. These samples indicate paleo RSL positions of $-0.53 \pm 0.25$ and $-0.11 \pm 0.25$ m. On Tambakulu, ages range between $114 \pm 114$ and $37 \pm 12$ years BP. In this time span, the elevations of the fossil microatolls at this island indicate RSL positions between $-0.24 \pm 0.13$ and $0.11 \pm 0.23$ m. The samples from Bone Batang cover ages from $5196 \pm 118$ to $3693 \pm 108$ years BP and provide paleo RSL positions of $0.13 \pm 0.22$ to $0.20 \pm 0.22$ m. Samples from fossil microatoll ages from Kodingareng Keke vary from $5869 \pm 99$ to $5343 \pm 88$ years BP, indicating paleo RSL positions between $-0.02 \pm 0.12$ and $0.10 \pm 0.12$ m. Fossil microatoll samples from Sanrobengi range in age from $5970 \pm 89$ to $3615 \pm 99$ years BP, with RSL from $0.14 \pm 0.12$ to $0.54 \pm 0.23$ m.

### 4.2   GIA models

As described in Sect. 3.5, we use different Earth and ice models to produce 54 different RSL predictions, from 16 ka to the present (Fig. 5b). The models are available in the form of NetCDF files including longitudes between 55.3 to 168.9° and latitudes between $-28.6$ and 38.6°. We provide the models in NetCDF format, with a Jupyter notebook to extract data at a single location and plot GIA maps (files can be retrieved from Rovere et al., 2020).

An extract of the modeling results is shown in Figs. 5 and 6. While all models predict an RSL highstand in the Spermonde Archipelago (Fig. 5a), the RSL histories predicted by each model show significant differences. ICE5g predicts the RSL highstand occurring ca. 2.5 kyr later than ANICE and ICE6g. The maximum RSL predicted by ICE5g

and ICE6g is higher than the one predicted by ANICE. ANICE is the only ice model for which at least one Earth model iteration (see the lowest line in Fig. 5) does not predict an RSL highstand but a quasi-monotonous sea level rise from 8 ka to the present.

## 5   Discussion

The dataset presented in Tables 2–4 and shown in Figs. 3a–c and 4 allow discussing several relevant points that need to be taken into account as sea-level studies in the Makassar Strait and SE Asia progress.

### 5.1   Measuring living microatolls for paleo RSL calculations

As indicated by previous studies (e.g., Mann et al., 2016; Smithers and Woodroffe, 2001; Woodroffe et al., 2012) the best practice to derive paleo RSL from microatolls is, when possible, to measure the HLC below MSL and use it to calculate their indicative meaning (Meltzner and Woodroffe, 2015).

Our results (Fig. 4) show that, in our study area, HLC is subject to changes over short spatial scales. In fact, within similar reef contexts, we measured significant differences in HLC across the Spermonde Archipelago, which seem to conform to a geographic trend directed from near the shore towards the islands located on the outer shelf. The highest HLC (i.e., closer to mean sea level) was measured at the island closest to the mainland (Sanrobengi). The islands located in the middle of the archipelago (Panambungan, Sanane, and Barrang Lompo) differ slightly from each other but show comparable average HLC. At Tambakulu, located further away from the mainland ($\sim 70$ km from Sanrobengi), the HLC is the lowest measured. On average, HLC at Tambakulu is $\sim 0.4$ m lower than that recorded at Sanrobengi. We highlight that this value is of the same magnitude (several decimeters) as the differences found by other studies reporting coral microatolls HLC measurements at different sites (Hallmann et al., 2018; Smithers and Woodroffe, 2001; Woodroffe, 2003; Woodroffe et al., 2012).

This pattern seems confirmed by the water level data we measured at the islands of Tambakulu and Barrang Lompo (Fig. 4c). While our measurements are too short in time to extract well-constrained tidal datums, we remark that at Tambakulu (offshore) we measured a tidal range that was higher than at Barrang Lompo (mid-shelf), which in turn records a slightly higher tidal range than the Makassar tide gauge (onshore). The local tidal range is related to the bathymetry and can, therefore, differ even in relative proximity. We highlight that, while a complete analysis of the water level data we surveyed is beyond the scope of this work, the PANGAEA dataset associated with this paper contains all the water levels recorded during our surveys for further analysis.

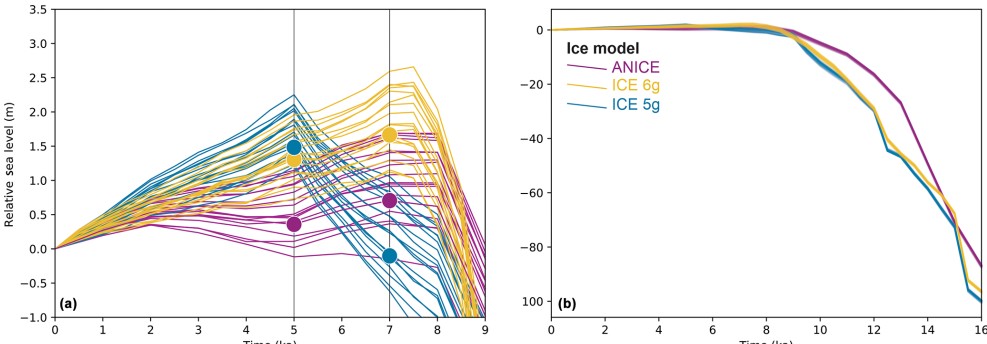

**Figure 5.** Results of the 54 GIA model runs for an island located in the center of the Spermonde Archipelago: **(a)** last 9 kyr – dots indicate the points at which the maps in Fig. 6 have been extracted; **(b)** last 16 kyr, representing the full time extent of the models. The eustatic sea level for each ice melting scenario is available in Rovere et al. (2020). The Jupyter notebook used to create this graph is available in Rovere et al. (2020).

**Table 3.** Fossil microatolls sampled by Mann et al. (2016) surveyed on Barrang Lompo (FMA 1 (BL)–FMA 7 (BL)) and Panambungan (FMA 8 (PPB)–FMA 21 (PPB)). All ages are recalculated with a $\delta R$ value of 0 and an error of 0 (Southon et al., 2002). All erosion corrections are already included in the RSL as provided in Mann et al. (2016), but all details are provided in the PANGAEA dataset associated with this paper. The elevation–age plot of these data is shown in Fig. 3a.

| Lab code | Sample name | Island name | $^{14}$C age | $\pm^{14}$C error | Mean age (cal yr BP) | $\pm$ Error (yr) | Elevation (m) with respect to m.s.l. | HLC (m) | RSL (m) | $\pm$ Vertical error (m) |
|---|---|---|---|---|---|---|---|---|---|---|
| Poz-63504 | FMA 1 (BL) | Barrang Lompo | 4505 | 30 | 4701 | 108 | −1.35 | −0.47 | −0.86 | 0.11 |
| Poz-66838 | FMA 2 (BL) | Barrang Lompo | 5600 | 40 | 6006.5 | 112.5 | −0.93 | −0.47 | −0.44 | 0.11 |
| Poz-63505 | FMA 3 (BL) | Barrang Lompo | 4405 | 35 | 4562 | 136 | −0.95 | −0.47 | −0.46 | 0.11 |
| Poz-66839 | FMA 4 (BL) | Barrang Lompo | 4900 | 35 | 5187 | 121 | −1.03 | −0.47 | −0.55 | 0.11 |
| Poz-63506 | FMA 5 (BL) | Barrang Lompo | 4965 | 35 | 5335 | 99 | −1.10 | −0.47 | −0.62 | 0.11 |
| Poz-66840 | FMA 6 (BL) | Barrang Lompo | 4640 | 35 | 4878 | 83 | −1.16 | −0.47 | −0.68 | 0.11 |
| Poz-66842 | FMA 7 (BL) | Barrang Lompo | 4830 | 40 | 5125 | 142 | −1.07 | −0.47 | −0.58 | 0.11 |
| Poz-66843 | FMA 8 (PP) | Panambungan | 5370 | 35 | 5746.5 | 109.5 | −0.30 | −0.50 | 0.39 | 0.13 |
| Poz-66844 | FMA 9 (PP) | Panambungan | 5185 | 35 | 5537.5 | 78.5 | −0.29 | −0.50 | 0.40 | 0.13 |
| Poz-66845 | FMA 10 (PP) | Panambungan | 5165 | 35 | 5521 | 72 | −0.27 | −0.50 | 0.42 | 0.13 |
| Poz-63507 | FMA 11 (PP) | Panambungan | 5325 | 35 | 5686 | 101 | −0.26 | −0.50 | 0.43 | 0.13 |
| Poz-63511 | FMA 12 (PP) | Panambungan | 4915 | 35 | 5193 | 131 | −0.38 | −0.50 | 0.11 | 0.11 |
| Poz-66846 | FMA 13 (PP) | Panambungan | 4940 | 40 | 5278 | 150 | −0.29 | −0.50 | 0.20 | 0.11 |
| Poz-63512 | FMA 14 (PP) | Panambungan | 3920 | 30 | 3905 | 100 | −0.50 | −0.50 | −0.02 | 0.11 |
| Poz-63513 | FMA 15 (PP) | Panambungan | 4645 | 30 | 4879 | 75 | −0.44 | −0.50 | 0.05 | 0.11 |
| Poz-66847 | FMA 16 (PP) | Panambungan | 4340 | 30 | 4479 | 88 | −0.47 | −0.50 | 0.02 | 0.11 |
| Poz-66848 | FMA 17 (PP) | Panambungan | 4330 | 35 | 4466.5 | 103.5 | −0.49 | −0.50 | −0.01 | 0.11 |
| Poz-66849 | FMA 18 (PP) | Panambungan | 4810 | 40 | 5106.5 | 149.5 | −0.44 | −0.50 | 0.05 | 0.11 |
| Poz-63515 | FMA 19 (PP) | Panambungan | 4940 | 35 | 5279 | 146 | −0.33 | −0.50 | 0.16 | 0.11 |
| Poz-66850 | FMA 20 (PP) | Panambungan | 5350 | 40 | 5724 | 118 | −0.34 | −0.50 | 0.35 | 0.13 |
| Poz-66852 | FMA 21 (PP) | Panambungan | 106.08 | 0.33 pMC | | | −0.44 | −0.50 | 0.04 | 0.11 |

The results discussed above stress the importance of measuring the HLC of living microatolls also at very small spatial scales. Had we only focused on the HLC published by Mann et al. (2016) for Panambungan, Sanane, and Barrang Lompo (located in the center of the archipelago), our paleo RSL reconstructions would have been biased. Specifically, we would have overestimated paleo RSL at Tambakulu and underestimated it at Sanrobengi. Our reconstructions would have been similarly biased had we, for our paleo RSL recon-

structions, used tidal datums derived from the tide gauge of Makassar.

## 5.2 Conflicting sea-level histories

Additionally to our new dataset and that of Mann et al. (2016) presenting index points, there are two studies reporting paleo sea-level observations for the Spermonde Archipelago: De Klerk (1982) and Tjia et al. (1972) (Fig. 7). Mann et

**Table 4.** Marine and terrestrial limiting indicators from De Klerk (1982) and Tjia et al. (1972) studied in different locations in SW Sulawesi and the Spermonde Archipelago. This table is an extract from the database by Mann et al. (2019b). The elevation–age plot of these data is shown in Fig. 3c.

| Lab code | Sample Name | Island name | $^{14}$C age | $\pm^{14}$C error | Mean age (cal yr BP) | $\pm$ Error (yr) | Elevation (m) with respect to m.s.l. | HLC (m) | RSL (m) | $\pm$ Vertical error (m) |
|---|---|---|---|---|---|---|---|---|---|---|
| GrN-9883 | – | Tanah Keke | 4165 | 64 | 4237 | 180 | 1.025 | NA | NA | NA |
| GrN-9884 | – | O. Pepe | 4260 | 64 | 4349.5 | 186.5 | 1.125 | NA | NA | NA |
| GrN-9885 | – | Talakaya | 2755 | 126 | 2503 | 189 | 1.22 | NA | NA | NA |
| GrN-10559 | – | Puntondo | 1525 | 130 | 1086.5 | 169.5 | 1.565 | NA | NA | NA |
| GrN-10560 | – | Puntondo | 1840 | 136 | 1410 | 189 | 1.84 | NA | NA | NA |
| GrN-10561 | – | Puntondo | 6540 | 103 | 7026.5 | 238.5 | 0 | NA | NA | NA |
| GrN-10562 | – | Puntondo | 4380 | 128 | 4562 | 230 | 1.365 | NA | NA | NA |
| GrN-10563 | – | Pamaroang | 4520 | 141 | 4689.5 | 257.5 | 1.825 | NA | NA | NA |
| GrN-10564 | – | Pangalasak | 2230 | 136 | 1828 | 232 | 1.25 | NA | NA | NA |
| GrN-10565 | – | Patene | 2330 | 136 | 1948 | 240 | 1.675 | NA | NA | NA |
| GrN-10566 | – | Samalona | 5440 | 150 | 5831 | 251 | 0 | NA | NA | NA |
| GrN-10491 | – | Tekolabua | 905 | 50 | 827.5 | 98.5 | 1.1 | NA | NA | NA |
| GrN-10492 | – | Tekolabua | 6840 | 100 | 7719 | 207 | −0.6 | NA | NA | NA |
| GrN-10493 | – | Maros | 6175 | 103 | 6624.5 | 243.5 | −0.5 | NA | −0.93 | 0.44 |
| GrN-10976 | – | Bone Tambung | 1735 | 83 | 1301.5 | 185.5 | 1 | NA | NA | NA |
| GrN-10978 | – | Sarappo | 3870 | 99 | 3837 | 267 | 0.7 | NA | NA | NA |
| GrN-10979 | – | Pamaroang | 3770 | 92 | 3709.5 | 240.5 | 3.56 | NA | NA | NA |
| GrN-10980 | – | Tarallow | 5740 | 106 | 6134.5 | 225.5 | 1.9 | NA | NA | NA |
| GrN-10981 | – | Puntondo | 8220 | 100 | 8738.5 | 261.5 | 1.53 | NA | NA | NA |
| GaK 3602* | – | Pamaroang | 4460 | 139 | 4610 | 372 | 5.75 | NA | NA | NA |
| GaK 3603* | – | Pamaroang | 5312 | 139 | 5656 | 323 | 6.5 | NA | NA | NA |

* Indicates samples from Tjia et al. (1972). NA: not available.

al. (2019a, b) reanalyzed data from these studies and recognized that most of the data originally interpreted as index points were instead better described as marine or terrestrial limiting indicators (Fig. 3c). Our new data agree with those from Mann et al. (2016) but show relevant differences with Tjia et al. (1972) and De Klerk (1982) studies, which place RSL at 6–4 ka conspicuously higher than what is calculated using the microatoll record (Fig. 3c).

This mismatch was recently pointed out by Mann et al. (2019a), who wrote "site-specific discrepancies between ... Tjia et al. (1972) ... and De Klerk (1982) and Mann et al. (2016) ... must be resolved with additional high-accuracy RSL data before the existing datasets can be used to decipher regional driving processes of Holocene RSL change within SE Asia".

While the study by Mann et al. (2016) was based only on two islands, the data presented in this study provide definitive evidence to call for a reconsideration of the data reported by Tjia et al. (1972) and De Klerk (1982). Notwithstanding the importance of these datasets, we highlight that the higher Late Holocene RSL histories reported by these two authors are largely at odds with more precise RSL indicators reported here. Hence, the following question arises: what is the possible reason for the Tjia et al. (1972) and De Klerk (1982) data

to be higher than the data reported by this study and Mann et al. (2016)?

One possible source of mismatch could reside in regional GIA differences. We suggest rejecting this hypothesis comparing the location of the areas surveyed in the Spermonde Archipelago with the outputs of our GIA models. Using the GIA models producing the most extreme differences within our region, we show that the discrepancy between the data cannot be explained by regional differences in the GIA signal. GIA differences remain within 1 m among our sites (Fig. 7a, b).

Similarly to GIA, another possible hypothesis is that the differences among sites in the Spermonde Archipelago are caused by differential tectonic histories between sites. While this is a possibility that would need further paleo RSL data to be explored (expanding the search of RSL indicators beyond the islands of the Spermonde Archipelago), we argue that there are several inconsistencies between the microatoll data and other sea-level data points surveyed within short geographic distances. For example, a fossil coral (not specified if in growth position) surveyed at Tanah Keke (GrN-9883, Table 4) by De Klerk (1982) would indicate that at $4237 \pm 180$ years BP, RSL was above 1.03 m. At the same time, microatoll data from Sanrobengi (SB_FMA25, Ta-

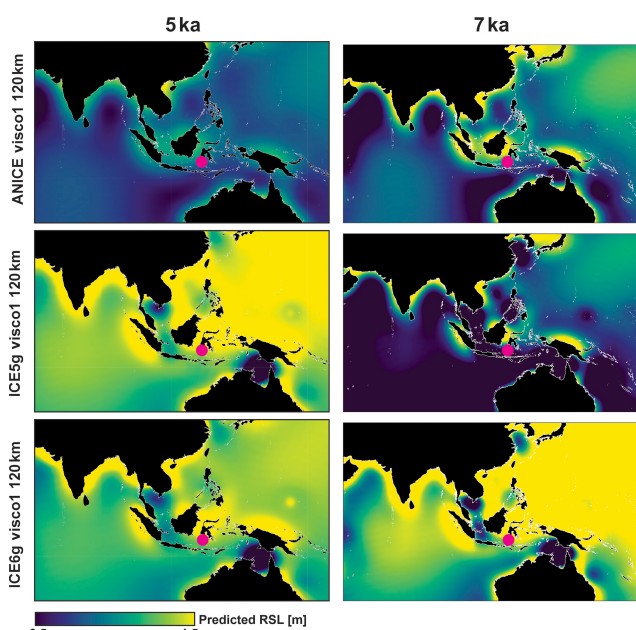

**Figure 6.** Relative sea level at 5 ka (left) and 7 ka (right) as predicted by three of the GIA models used in this study. See Table 1 for the definition of the mantle viscosity here labeled "Visco1". The purple dot indicates the approximate position of the Spermonde Archipelago.

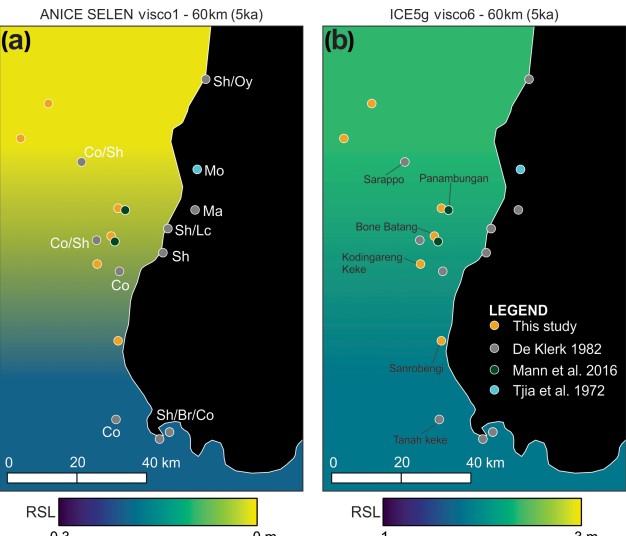

**Figure 7.** Location of the RSL data presented in this study, Mann et al. (2016), De Klerk (1982), and Tjia et al. (1972) compared with RSL as predicted by GIA models. Land areas are filled in black. Here we show the models predicting the lowest **(a)** and highest **(b)** RSL in the Spermonde Archipelago. Labels in **(a)** represent the type of indicator reported by De Klerk (1982) and Tjia et al. (1972). Island names in **(b)** refer to the islands mentioned in the discussion. Legend: Sh – shell accumulations; Oy – Oysters (no further details available); Mo – mollusks fixed on Eocene bedrock; Ma – mangrove swamp; Lc – loamy clays; Br – beachrock; Co – corals (in situ?). In **(b)** we report the names of the islands discussed in the main text.

ble 2, ∼ 20 km north of Tanah Keke) show that RSL was 0.46 ± 0.23 m above present sea level. Similarly, at the site of Sarappo, De Klerk (1982) surveyed coral and shell accumulations that would propose the sea level was above 0.7 m at 3837 ± 267 years BP (GrN-10978). This data point is at odds with microatoll data from the nearby islands of Panambungan, Bone Batang, and Sanrobengi, where, at the same time, RSL is recorded by microatolls at elevations between −0.02 ± 0.11 and 0.46 ± 0.23 m (BB_FMA13, SB_FMA26, Table 2 and FMA14 (PP), Table 3). We argue that invoking significant differential tectonic shifts between islands located so closely in space would require the presence of tectonic structures on the shelf of the Spermonde Archipelago that are, at present, unknown.

Another possibility is that, while the original descriptions of Tjia et al. (1972) and De Klerk (1982) seem to indicate "marine limiting" points (i.e., indicating that sea level was above the measured elevation; Mann et al., 2019a), some of them may instead be representative of terrestrial environments and thus naturally above our paleo RSL index points. For example, it is not clear whether the "shell accumulations" reported at several sites and interpreted by Mann et al. (2019a) as marine limiting points may be instead representative of high-magnitude wave deposits by storms. The Spermonde Archipelago is subject to occasional strong storms that may explain the high emplacement of these deposits (see wave statistics in Fig. 8).

Also, tsunamis are not unusual along the coasts of SE Asia (e.g., Rhodes et al., 2011) with the broader region in the Makassar Strait being one of the most tsunamigenic regions in Indonesia (Harris and Major, 2017; Prasetya et al., 2001). Nevertheless, the tsunamigenic earthquakes reported in this region are far north of our study area (Prasetya et al., 2001; see the left panel in Fig. 8), and in general, they appear shallow and too small in magnitude to produce significant tsunamis propagating towards the Spermonde Archipelago. The earthquakes in this area are all generated along the Paternoster transform fault, which would point to tsunamis generated mostly by earthquake-triggered landslides rather than earthquakes themselves. Nevertheless, a tsunamigenic source for marine sediment deposition significantly above MSL cannot be ruled out until the deposits reported by Tjia et al. (1972) and De Klerk (1982) are reinvestigated with respect to their precise elevations above MSL and their sediment facies.

Only further field data at the locations reported by Tjia et al. (1972) and De Klerk (1982) might help clarify the stratigraphy of these deposits and the processes that led to their deposition (i.e., paleo sea-level changes versus high-energy events).

Maximum significant wave height 1979–2016 [m]     Maximum significant wave period 1979–2016 [s]

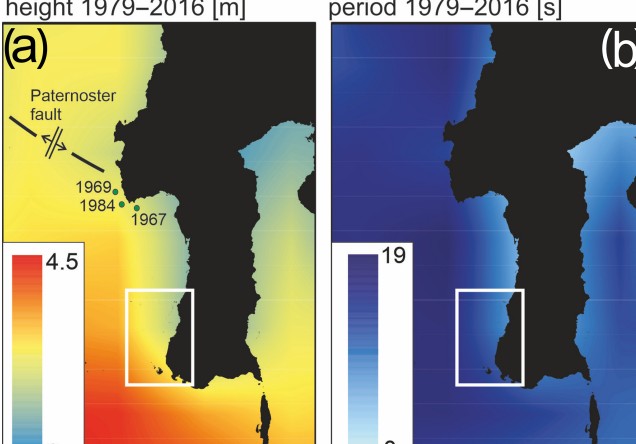

**Figure 8.** Maximum significant wave height **(a)** and period **(b)** extracted from the CAWCR wave hindcast (Durrant et al., 2013, 2015). Panel **(a)** shows the approximate location and year of the three historical tsunami records reported by Prasetya et al. (2001); their Fig. 1. Faultline and axis of spreading of the Paternoster fault are derived from Prasetya et al. (2001); their Fig. 5. The box delimited by the white line indicates the approximate location of Fig. 7 within this figure. CAWCR source: Bureau of Meteorology and CSIRO Copyright 2013.

### 5.3    Subsidence at a highly populated island?

As shown in Fig. 3a, the data presented in this study together with the data from Mann et al. (2016) confirm a sea-level history with a higher-than-present RSL at 6–3.5 ka. The only exception to this pattern is the island of Barrang Lompo, where microatolls of roughly the same age are consistently lower (light blue crosses in Fig. 3a). CE1 We compare the data at Barrang Lompo to the other RSL data points in the Spermonde Archipelago using a Monte Carlo simulation (see Rovere et al., 2020 for details and methods) to highlight spatiotemporal clustering in these two datasets. We calculate that, on average, at $\sim 5100$ years BP, RSL at Barrang Lompo is $0.8 \pm 0.3$ m lower than all the other islands where we surveyed microatolls of the same age (Fig. 9).

The mismatch in RSL histories shown above can hardly be reconciled by differential crustal movements due to either tectonics or GIA over such short spatial scales (Fig. 1b). For example, Bone Batang (where fossil microatolls were surveyed slightly above present sea level) and Barrang Lompo (where microatolls of roughly the same age were surveyed ca. 0.8 m below those of Bone Batang) are separated by less than 5 km, and it is, hence, highly unlikely that they were subject to very different tectonic or isostatic histories. The only geographic characteristic that separates Barrang Lompo from the other islands we surveyed is that it is heavily populated ($\sim 4.5$ thousand people living on an island of $0.26 \, \text{km}^2$) (Syamsir et al., 2019). As such, it is characterized by a very

dense network of buildings and concrete docks. The island is also subject to groundwater extraction (at least eight wells were reported on Barrang Lompo; Syamsir et al., 2019).

The island of Barrang Lompo has been populated since at least the 1720s (Clark, 2013; de Radermacher, 1786 as cited in Schwerdtner Máñez and Ferse, 2010) when Barrang Lompo was (as it is today) a hub for sea cucumber fisheries (Schwerdtner Máñez and Ferse, 2010). Assuming that the localized subsidence is anthropogenic, we cannot exclude that it started since the early colonization, but it seems appropriate to date it back to, at least, 100–150 years ago. At this time, the island population likely started to grow and to extract more groundwater for its sustenance. Using these inferences, our microatoll data show that Barrang Lompo might be affected by a subsidence rate in the order of $\sim 3$–$11 \, \text{mm} \, \text{yr}^{-1}$ (depending on the adopted subsidence amount and time of colonization) compared to the non-populated islands in the archipelago. Notwithstanding the obvious differences in patterns and causes of subsidence, we note that this rate is at least 1 order of magnitude smaller than what is observed in Indonesian megacities due to anthropogenic influences (Alimuddin et al., 2013). As this subsidence rate is a relative rate among different islands, any other regional subsidence or uplift rate (i.e., tectonic uplift or GIA-induced vertical land motions) should be added to this estimate.

As the fossil microatolls surveyed at anomalous positions were all located near the coast, one possibility is that they might have been affected by local subsidence due to the combined effect of groundwater extraction and construction load on the coral island. One point worth highlighting is that the depth of living microatolls, surveyed on the modern reef flat a few hundred meters away from the island, does not show significant differences when compared to other islands nearby (Fig. 4b). If the island is indeed subsiding, this observation could be interpreted in two ways. One is that the subsidence might be limited to the portions closer to the shoreline and not to the distal parts (i.e., the reef flat) where modern microatolls are growing. The second is that the island has been subsiding fast in the recent past but is now subsiding at roughly the same rate of upward growth of the living microatolls (Simons et al., 2007). Meltzner and Woodroffe (2015) report that microatolls are in general characterized by growth rates of $\sim 10 \, \text{mm} \, \text{yr}^{-1}$, with extremes between 5 to $25 \, \text{mm} \, \text{yr}^{-1}$ for those belonging to the genus *Porites*. These rates would allow modern microatolls to keep up with relative sea-level rise (which includes subsidence) at Barrang Lompo. We remark that, on average, living microatolls at Barrang Lompo are slightly thicker than those of islands nearby (Mann et al., 2016, Fig. 4a).

A partial indication of a possible subsidence pattern at Barrang Lompo is given by the intense erosion problems that this island is reported to experience, which may be the consequence of high rates of land subsidence. Relatively recent reports indicate that coastal erosion is a particularly striking problem at Barrang Lompo (Williams, 2013; Tahir et

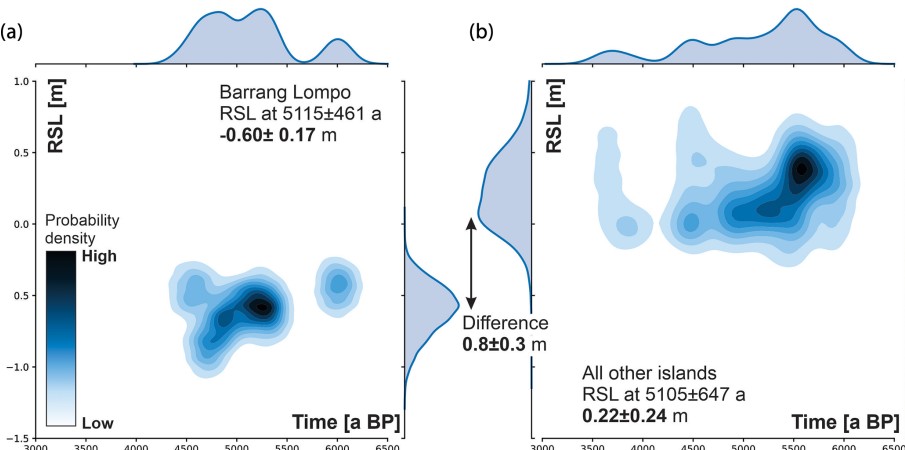

**Figure 9.** Joint plot showing bivariate (central plot) and univariate (marginal axes) distribution of RSL data points at Barrang Lompo **(a)** and all the other islands surveyed in this study and in Mann et al. (2016) **(b)**. Darker blue areas in the central plots indicate a higher density of RSL point therefore darker colors indicate a higher probability of RSL at the given time. The Jupyter notebook used to create this graph is available in Rovere et al. (2020).

al., 2009). Tahir (2009) indicates that large parts of the island suffer from severe erosion problems. Based on community interviews, they conclude that the coastline may have retreated at a pace of $0.5\,\mathrm{m\,yr^{-1}}$. Williams (2013) reported that "local people had constructed a double seawall of dead coral to mitigate erosion".

We recognize that the mechanism of subsidence for Barrang Lompo proposed above should be regarded as merely hypothetical and needs confirmation through independent datasets. For example, the RSL change rates we propose for Barrang Lompo would be observable by instrumental means. A comparative study using GPS measurements for a few days per year for 3–5 years would provide enough information to inform on vertical land motion rates in Barrang Lompo. Another approach would be the use of tide gauges to investigate multiannual patterns of land and sea-level changes in Barrang Lompo and at other populated and non-populated nearby islands. This would surely help to understand the reasons for the mismatch highlighted by our data.

Another way to detect recent vertical land movements between the island of Barrang Lompo and other uninhabited islands nearby would be to investigate whether there are differences in the morphology and growth patterns of living microatolls. If Barrang Lompo's rapid subsidence is also affecting also the distal part of the reef, this may be detectable through higher annual growth rates of the microatolls at this island compared to that measured at other islands.

To our knowledge, there is only one instrumental example of the kind of subsidence we infer here. At Funafuti Island (Tuvalu), Church et al. (2006) report that two closely located tide gauges (ca. 3 km apart) show a difference in RSL rise rates. In the search for an explanation to this pattern, they infer that "this tilting may be caused by tectonic movement or (most probably) local subsidence (for example, due to groundwater withdrawal) and demonstrates that even on a single island, the relative sea-level trend may differ by as much as $0.6\,\mathrm{mm\,yr^{-1}}$".

## 5.4 Common Era microatolls

Eight microatolls from the islands of Suranti and Tambakulu (located in the north of our study area, 12 km apart from each other) yielded ages spanning the last $\sim 300$ years (Fig. 3b). This period represents the most recent part of the Common Era. Sea-level data from this period are relevant to assess rates of sea-level changes beyond the instrumental record (Kopp et al., 2016). Within Southeast Asia, the database of Mann et al. (2019b; https://doi.org/10.17632/mr247yy42x.1) reports only one index point for this time frame (Singapore; Bird et al., 2010).

As the two islands of Suranti and Tambakulu are uninhabited and hence are not subject to the hypothetical anthropogenic subsidence discussed above for the island of Barrang Lompo, it is possible to use these data to calculate short-term vertical land motions. To do this, we first need to correct the paleo RSL as reported in Fig. 3b to account for the 20th-century sea-level rise and GIA land uplift since the microatolls were drowned (see Rovere et al., 2020, for the complete calculation). We make this correction using the 20th-century global sea-level rise of $184.8 \pm 25.9\,\mathrm{mm}$ (Dangendorf et al., 2019) and GIA rates from our models ($0.38 \pm 0.09\,\mathrm{mm\,yr^{-1}}$; see the PANGAEA dataset associated with this paper for details). We then iterate multiple linear fits through our data points by randomly selecting ages and CE RSL corrected as described above (full procedure and script available in Rovere et al., 2020). After $10^4$ iterations, we calculate that the average vertical land motion (VLM) rate indicated by our microatolls is $-0.88 \pm 0.61\,\mathrm{mm\,yr^{-1}}$ (Fig. 10).

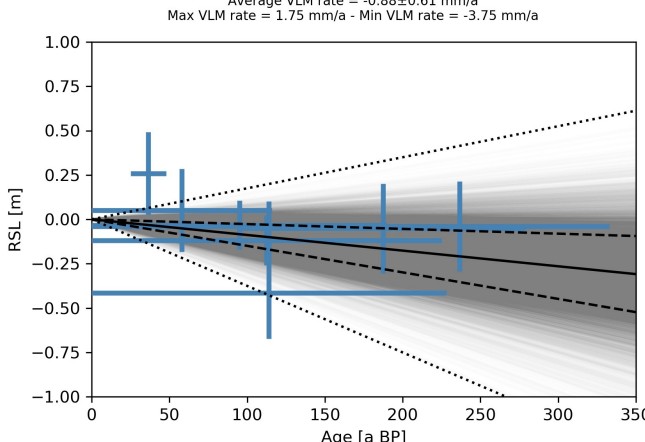

**Figure 10.** Common Era data points, corrected for 20th-century sea-level rise and GIA uplift (blue crosses). Gray lines show the results of reiterating a linear fit through random normal samples of the blue points. Dotted black lines show the linear fits with maximum and minimum slopes. Dashed black lines show average + standard deviation and average − standard deviation slopes. The solid black line shows the average slope. The Jupyter notebook used to create this graph is available in Rovere et al. (2020).

While this range indicates that natural subsidence might be occurring at these islands, we cannot discard the possibility of a slight uplift, or stability.

We recognize that the calculation applied above to our data represents an approximation. Hence, the calculated rate is subject to several sources of uncertainty. First, five of eight Common Era microatolls were eroded; therefore, the paleo RSL might be overestimated. Second, four of eight microatolls have large age error bars. Then, in our calculations, we use global mean sea-level rise rates instead of local ones, which are not available for this area. The GIA models we employ are also limited, although they span a large range of possible mantle and ice configurations. Yet our calculation is the best possible with the available data.

Notwithstanding the caveats above, we observe that the vertical land motion rates we calculate based on Common Era microatolls ($-0.88 \pm 0.61\,\mathrm{mm\,yr^{-1}}$) are in agreement with the average vertical motion of $-0.92 \pm 0.53\,\mathrm{mm\,yr^{-1}}$ reported by Simons et al. (2007) (see Table S6 in the their Supplement) for the *PARE* GPS station (lat $-3.978°$, long $119.650°$; height: 135 m). This station is located on the mainland, 78 km ENE of Tambakulu and Suranti. Nevertheless, the subsidence indicated by both our data and the *PARE* station appears to be at odds with another GPS station reported by Simons et al. (2007) in the proximity of Makassar (*UJPD*; lat $-5.154°$, long $119.581°$; height: 153 m), which instead measures uplift rates of $2.78 \pm 0.60\,\mathrm{mm\,yr^{-1}}$. While caution is needed when comparing long-term rates to the short-term ones measured by GNSS stations, these results provide important stepping stones for future studies in this area.

## 5.5 Comparison with GIA models

Excluding the microatoll data from the island of Barrang Lompo (which, as per the discussion above, may have been subject to recent subsidence), 29 fossil microatolls in the Spermonde Archipelago (including also the data reported by Mann et al., 2016, Fig. 3a) date to between 3615 and 5970 years BP. This dataset can be compared with the predicted RSL from GIA models once vertical land movements are considered. To estimate such movements in the Spermonde Archipelago, two options are available.

The first is to consider that the area has been tectonically stable during the Middle Holocene. This is plausible under the notion that, unlike the northern sector of western Sulawesi (which is characterized by active lateral and thrust faults; Bird, 2003), southern Sulawesi is not characterized by strong tectonic movements (Sasajima et al., 1980; Hall, 1997; Walpersdorf et al., 1998; Prasetya et al., 2001). Considering the Spermonde Archipelago as tectonically stable (Fig. 11a), our RSL data show the best fit with the RSL predicted by the ANICE model (VM2 – 60 km; see Table 1 for details), in particular with those iterations predicting RSL at 6–4 ka a few decimeters higher than present.

The second option is to interpret the rate of RSL change calculated from Common Era fossil microatolls ($-0.88 \pm 0.61\,\mathrm{mm\,yr^{-1}}$) and make two assumptions: (1) that they were uniform through time and (2) that they can be applied to the entire archipelago. Under these assumptions, we show in Fig. 11b that, with subsidence rates below $-0.5\,\mathrm{mm\,yr^{-1}}$, our data do not match any of our RSL predictions. Data start to match RSL predictions obtained using the ICE6g ice model with lower subsidence rates. For example, with a subsidence rate of $-0.27\,\mathrm{mm\,yr^{-1}}$, representing the upper end of the $2\sigma$ range shown in Fig. 10, the data show a good match with ICE6g (Fig. 11c). As discussed above, based on both our Common Era data and GPS data from Simons et al. (2007) we cannot exclude that, instead of subsidence, the archipelago is characterized by tectonic uplift. The maximum uplift compatible with our RSL data and models is $0.05\,\mathrm{mm\,yr^{-1}}$ (Fig. 11d). Regardless of the tectonic history chosen, we note that our data do not match the peak highstand predicted at 5 ka by the iterations of the ICE5g model.

## 5.6 Paleo to modern RSL changes

Due to the existing uncertainties on vertical land motions discussed above, it is clear that the data in the Spermonde Archipelago cannot be used to infer global mean sea level. Yet the matching exercise of our RSL data with GIA models under different vertical land motion scenarios shown in Fig. 11 allows discussing the contribution of GIA to relative sea-level changes at broader spatial scales. GIA effects need to be taken into account in the analysis of both tide gauge and satellite altimetry data (see Rovere et al., 2016, for a review). One way to choose the GIA model(s) employed for this cor-

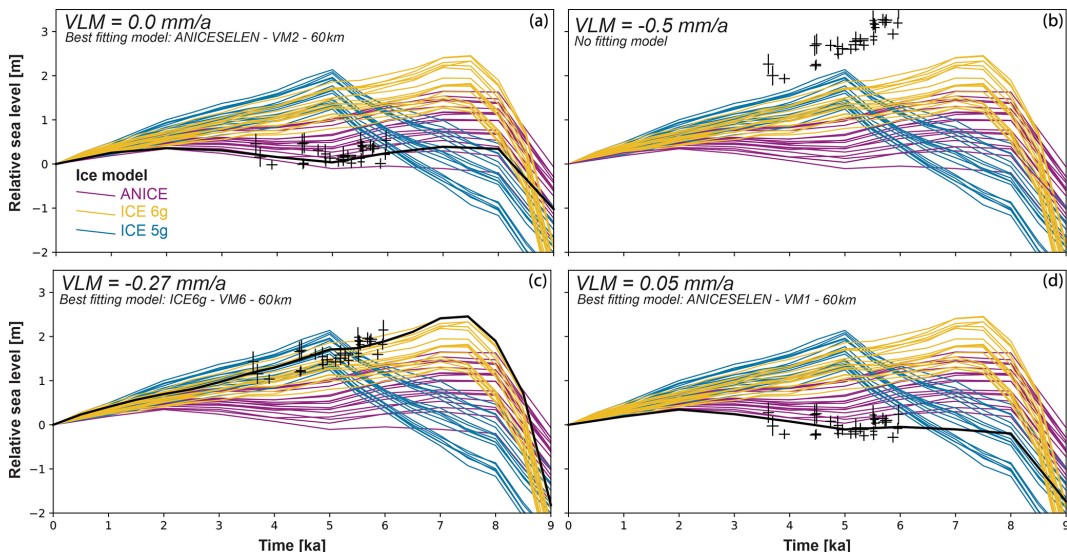

**Figure 11.** Comparison between RSL observations (except the island of Barrang Lompo) and predictions from GIA models (see Table 1 for model details). The model predictions were extracted by averaging latitude and longitude of all islands reported in this study, minus Barrang Lompo. Colored lines represent ANICE, ICE5g, and ICE6g models. Thicker, black lines identify the best fitting models. The different panels **(a–d)** show different tectonic corrections applied to the observed RSL data. The Jupyter notebook used to create this graph is available in Rovere et al. (2020).

rection is to select those matching better with Late Holocene data.

To make an example of how different modeling choices (based on RSL data) propagate onto estimated modern GIA rates, in Fig. 12a–c, CE2 we show the land motion rates caused by GIA as predicted by three models across southern and Southeast Asia. These are the broad geographic results associated with the best-matching models under different assumptions on VLM (as shown in Fig. 11). The difference between the two most extreme models matching with our data is within $-0.3$ and $0.5 \, \text{mm} \, \text{yr}^{-1}$ (Fig. 12d).

To give an example of the difference between these models, Fig. 12d shows that ICE6g-VM6-60km predicts faster modern CE5 GIA rates than ANICESELEN-VM1-60km for India and Sri Lanka. As these rates would need to be CE6 subtracted from the data recorded by a tide gauge, this would affect any attempt to decouple the magnitude of eustatic vs other land motions at tide gauges in that area.

## 6 Conclusions

In this study, we report 25 new RSL index points (of which one was rejected due to evidence of reworking) and 75 living microatoll measurements from the Spermonde Archipelago. We also report 54 new GIA model iterations that span a large geographic region extending beyond Southeast Asia. Together with the data reported in Mann et al. (2016), these represent an accurate dataset against which paleo RSL changes in the Spermonde Archipelago and adjacent coasts (including the city of Makassar, the seventh-largest in Indonesia)

can be benchmarked. Multiple implications are derived from our discussions. We summarize these below.

Our measurements of living microatolls show that there is an elevation difference between the HLC results from the nearshore islands of the archipelago (Sanrobengi, Fig. 1) towards the outer shelf ones (Tambakulu and Suranti, Fig. 1). The magnitude of this gradient or slope seems to be confirmed by water level data we measured at different islands and is ca. 0.4 m, with living microatolls deepening towards the offshore area. Recognizing the presence of this gradient was important to obtain coherent RSL reconstructions among different islands. This strengthens the notion that, when using microatolls as RSL indicators, living microatolls must be surveyed near fossil ones to avoid biases in sea-level reconstructions.

The data surveyed in the Spermonde Archipelago by De Klerk (1982) and Tjia et al. (1972) are largely at odds with precisely measured and interpreted fossil microatolls presented in this study. We propose that, pending more accurate elevation measurements and reinterpretation of these data, they are excluded from sea-level compilations (i.e., Mann et al., 2019b in Khan et al., 2019). We also propose that there is the possibility that these deposits might represent storm (or tsunami) accumulations: this hypothesis needs further field investigations to be tested.

Data from the heavily populated island of Barrang Lompo plot significantly lower (ca. 80 cm) than those at all the other islands. Here, we propose the hypothesis that groundwater extraction and loading of buildings on the island may be the cause of this discrepancy, which would result in local

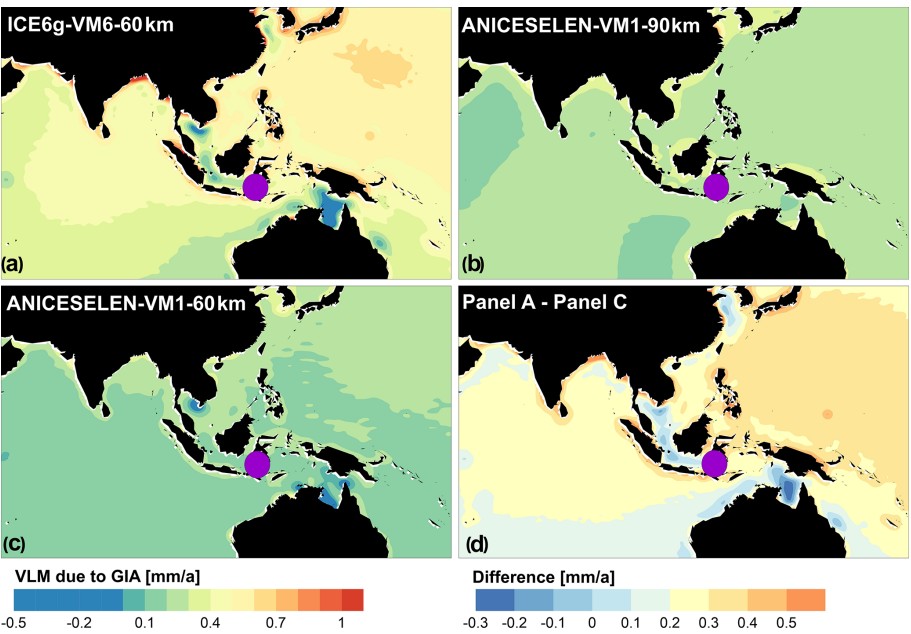

**Figure 12.** CE3 **(a–c)** CE4 GIA-induced vertical land motion derived by linearly interpolating the last time step in our models (1 kyr for ANICE, 0.5 kyr for ICE6g) to the present. **(d)** Difference between the models with the most extreme predictions matching our Late Holocene sea-level index points under different vertical land motion scenarios (see Fig. 11). The purple dot indicates the approximate position of the Spermonde Archipelago.

subsidence rates of Barrang Lompo in the order of $\sim 3$–$11\,\mathrm{mm\,yr^{-1}}$. Due to the lack of instrumental data to support our hypothesis, we highlight the need for future studies acquiring both instrumental records and high-resolution RSL histories from fossil microatolls (e.g., reconstructing die-downs from microatoll slabs) across islands with different human population patterns. This mechanism of local subsidence needs to be verified with independent data. If confirmed, this would have wider implications for the resilience of low-lying, highly populated tropical islands to changes in sea level.

Besides the mechanism of local anthropogenic subsidence which we propose for the island of Barrang Lompo, eight microatolls dating to the last ca. 300–400 years allow us to calculate recent vertical land motion rates. We calculate that our data may indicate average vertical land motion rates of $-0.88 \pm 0.61\,\mathrm{mm\,yr^{-1}}$. As these rates were calculated only for the two offshore islands in our dataset, we advise caution in extrapolating to broader areas. Nevertheless, we point out that this rate of subsidence is very consistent with that derived from a GPS station less than $100\,\mathrm{km}$ away (which recorded a rate of $-0.92 \pm 0.53\,\mathrm{mm\,yr^{-1}}$; Simons et al., 2007), but is at odds with another GPS station in Makassar, for which uplift is reported.

Comparing the part of our dataset dated to 3–4 ka with the RSL predictions from a large set of GIA models, we show that the best matching ice model depends on the assumptions on vertical land movements. A generally better fit with

models using the ICE6g ice history is obtained with moderate subsidence rates ($-0.27\,\mathrm{mm\,yr^{-1}}$), while models using the ANICE ice history are more consistent with hypotheses of stability or slight tectonic uplift ($0.05\,\mathrm{mm\,yr^{-1}}$). The ice model ICE5g shows a peak in RSL at ca. 5 ka that does not match our RSL observations at the same time.

In this study, we are not favoring one model over the others nor claim that our model ensemble is a complete representation of the possible variable space. We use the example of the Spermonde Archipelago to highlight how Holocene RSL data, coupled with GIA models, can inform on two aspects that are ultimately of interest to coastal populations.

First, they may help to benchmark subsidence rates obtained from GPS or tide gauges. It appears that, for the Spermonde Archipelago, long-term subsidence, tectonic stability, or slight uplift are all possible. To settle this uncertainty, instrumental measures and more precise Common Era sea-level datasets should represent a focus of future sea-level research in this area.

Second, we showed here that matching GIA model predictions with Late Holocene RSL data is useful to constrain which models might be a better choice to predict ongoing regional rates of GIA. While we do not have a definite "best match" for the Spermonde Archipelago, we suggest that iterations of ICE6g and ANICESELEN fit better with our data and might produce more reliable GIA predictions than ICE5g, which seems not to match our data as well as the other two. To enable data–model comparisons such as the

one performed in this study an online repository (Rovere et al., 2020) contains all our model results at broad spatial scales for southern and Southeast Asia.

**Data availability.** The original data presented in this paper are available in PANGAEA under the link: TS1. The dataset includes a spreadsheet containing (1) site coordinates; (2) water level logger data; (3) details of MSL calculations; (4) complete data tables (also including the data reported in Tables 2, 3, and 4 in the main text); (5) data for each island, as collected in this study or reassessed from Mann et al. (2016); (6) data on modern GIA rates; (7) results of XRD elemental analysis; and (8) details on living microatolls. NetCDF files of GIA models and a collection of Jupyter notebooks to reproduce the analyses in the paper are available here: https://doi.org/10.5281/zenodo.3593965 (cited as Rovere et al., 2020 in the text).

**Supplement.** The supplement related to this article contains the original radiocarbon laboratory reports and is available online at The supplement related to this article is available online at: https://doi.org/10.5194/cp-16-1-2020-supplement.

**Author contributions.** MB organized fieldwork and sampling, which were conducted in collaboration with TM and DK. JJ gave on-site support in Makassar and provided essential support with sampling and research permits in Indonesia. MB organized the data analysis, with supervision and inputs by TM and AR. The python codes provided in the Supplement were written by AR, in collaboration with MB and PS. TS and JI analyzed the tidal datum and calculated MSL, providing expertise on modern sea-level processes. PS offered expertise, performed model runs and provided discussion inputs on glacial isostatic adjustment. MB drafted the first version of the paper. MB and AR wrote the final version of the paper jointly. All authors revised and approved the content of the paper.

**Competing interests.** The authors declare that they have no conflict of interest.

**Acknowledgements.** We would like to thank SEASCHANGE (RO-5245/1-1) and HAnsea (MA-6967/2-1) from the Deutsche Forschungsgemeinschaft (DFG), which are part of the Special Priority Program (SPP)-1889 "Regional Sea Level Change and Society" for supporting this work. Thanks to Thomas Lorscheid and Deirdre Ryan for help and thoughtful comments. We acknowledge the help of the following Indonesian students and collaborators: Andi Eka Puji Pratiwi "Wiwi", Supardi, and Veronica Lepong Purara, who provided support during fieldwork and sampling. We are grateful to the Indonesian Ministry for Research, Technology and Higher Education (RISTEKDIKTI) for assistance in obtaining research permits. The fieldwork for this study was conducted under Research Permit No. 311/SIP/FRP/E5/Dit.KI/IX/2017. We are also grateful to the Badan Informasi Geospasial (BIG), Indonesia, for sharing Makassar tide gauge data.

**Financial support.** This research has been supported by the Deutsche Forschungsgemeinschaft (SEASCHANGE (grant no. RO-5245/1-1)).

The article processing charges for this open-access publication were covered by the University of Bremen.

**Review statement.** This paper was edited by Pierre Francus and reviewed by Daria Nikitina and three anonymous referees.

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

## Remarks from the language copy-editor

**CE1** Unfortunately, we cannot insert text at this point without consulting the editor for approval. Please provide a clarification for the editor as to why this insertion is necessary. In a few other cases, the requested change also appears to change the meaning in a substantial way, and as we cannot judge these changes in terms of content, we are also required to obtain the editor's approval in such cases. I have therefore inserted a few more blackboxes requesting clarification from the editor. Thank you in advance!

## Remarks from the typesetter