# Peer review of "Late Holocene (0-6ka) sea-level changes in the Makassar Strait, Indonesia"

_Climate of the Past, 2019_

## Referee Comment (RC1) · Anonymous Referee #1 · 26 Jun 2019

This paper develops 24 new sea-level index points using radiocarbon-dated coral microatolls from South East Sulawesi in Indonesia. The dates span the second half of the Holocene (past 6 ka) and capture a time period where Earth-ice models predict a highstand in sea level. To be suitable for publication in Climate of the Past (or elsewhere) I believe that a wholesale re-evaluation of the manuscript and underlying data are necessary.

(1) What is the purpose of this study?

Unfortunately, this paper is largely a description of some new data. The reader is not provided with a specific and compelling reason for why the work was undertaken in the first place, or why the study sites and region are important. Furthermore, there is no explicit take home message about the wider implications of the results. As such, the

current manuscript is only of local interest to others studying coral microatolls and sea level near Makassar, Indonesia.

In the introduction the authors should provide an explicit motivation for their work. Please explain (1) why is it necessary to document the height and timing of a Holocene sea level highstand given that it has been done in so many other places already? (2) What is the significance of SE Sulawesi? What can be learned by reconstructing relative sea level here that wasn't already known from existing studies? The abstract (lines 38-40) and introduction (lines 55-61) frame this study around the threat of future sea-level rise. However, the paper makes no effort to use the results in this context (see conclusions for example). Future sea-level rise is a convenient angle for making the topic of a paper appear widely relevant, but unless the paper leverages paleo results to produce better predictions this motivation should be removed.

(2) Mechanisms of Subsidence (section 5.4)

One of the five sites (Barung Lompo) records relative sea level that is anomalously low (Figure 3E). The authors propose that the reason for this trend is subsidence and go onto to propose that loading by approximately 4500 people living there, building concrete docks, and extracting water from wells is the driver. This explanation seems unfeasible to me. The amount of subsidence is large (order 1m per figures 3 and 7 and line 383), must be very fast (order of 1cm/year assuming that the development occurred in the last 100 years), and it doesn't seem unusual to me that this level of development would occur on a coral island in which case surely there would be lots of islands in many tropical places showing rapid subsidence at locations all around the world. Furthermore, the subsidence appears to be restricted spatially to just the island itself and doesn't extend onto the adjacent reef flat (line 398). Rather than offering an unsubstantiated idea, please could the authors provide some evidence in the form of (for example) loading calculations or discussion of instrumented examples from other inhabited coral islands that are subsiding.

Figure 7 is not necessary. Its very easy to see from figure 3 that the difference is about 1m. Why doesn't the peak in the smooth curve line up with the peak in the histogram?

(3) Common Era sea level (section 5.5)

The authors have 8 microatolls that formed in the last 400 years and these form the basis for this section of the discussion. I see no purpose for this discussion and was unable to see its relevance to the paper or wider discussions about Common Era sea level. The database used by Kopp (2016) is not an exhaustive list of all sea level index points, it states clearly that the data are limited to detailed records. There are likely 100s of sea-level index points that were NOT included in Kopp.

The authors claim that their new sea-level index points are the first from South East Asia for the Common Era because there is no data in Kopp (2016) from this region. This is simply not true and ignorance of the literature is an inadequate caveat. As one simple and widely known example see Horton et al. (2005; https://journals.sagepub.com/doi/10.1191/0959683605hl891rp). I expect the World Atlas of Holocene Sea-Level Curves would probably also provide some examples of Common Era sea-level index points from South East Asia.

Why is the Common Era even discussed? What is the importance of this time period (as distinct from the late Holocene which this paper focuses on), particularly given that the highstand occurs earlier? Figure 8C shows that the new index points have uncertainties that are too large to be useful in inferring much about Common Era sea level.

(4) Conflicting sea-level histories (section 5.1)

This section of the paper (as written) seemed unnecessary to me. The authors discuss the Tija (1972) and De Klerk (1982) sea level reconstructions, but explain that a paper by Mann et al. (under review) contradicts the original interpretations of the two older papers. Why does this paper need to summarize the arguments made in Mann et al?

[Figure]

To retain this section, the authors need to show the Tija (1972) and De Clerk (1982) data and provide a through explanation as to why those interpretations should be refuted. There are major contradictions in this section. Line 316 states that "Makassar Strait being one of the most tsunamigenic regions in Indonesia". Line 330 states that this study and others before it assumed that Makassar Strait was tectonically stable. How can the authors justify these two statements – they seem fundamentally opposed to one another. Is tectonics on or off the table for interpreting the new relative sea level curves?

I think that figure 3 shows the new sea level histories to be conflicting within the study area. The authors should focus on this rather than Mann's (under review) of old data. Although Barrang Lompo is identified as an anomalous site, there are others that don't match up particularly well with one another. For example, Bone Bafang (Figure 3F) seems to sit systematically lower than Sanrobengi (Figure 3B) by about 25-30 cm. Why is this the case? Its hard to evaluate the significance of this difference without knowing tidal range (not provided anywhere in the paper). Why is the relative sea level fall at Panambungan (Figure 3G) at 6 to 5 ka BP not seen anywhere else? None of these conflicting sea-level histories are discussed (and much less explained) by the paper. These discrepancies seem very large given how close the sites are to one another (within about 100 km) and until this variability is explained the authors cannot justify their later interpretations of regional trends in relation to GIA models or Common Era trends for example.

(5) Validation of GIA models (section 5.2)

This section of the paper offers minimal insight. The new reconstructions are compared to some GIA models. Unsurprisingly, there is a spread of highstand predictions (height and timing) among the different GIA models and therefore the agreement to the new data also varies among models. The authors proceed to identify models that fit the data better and worse. What should a reader take away from this section? Should we discard ICE5G because its fit to data in South East Sulawesi is worse than ANICE?

Why does the fit in this region offer more insight that the fit in other regions (particularly those with much bigger and longer datasets such as the British Isles or South East Asia, where databases on index points have long been compared to GIA predictions).

Line 344 states "The better match of ANICE to our data has a meaning for which concerns ice melting patterns. In fact, the lower highstand predicted from ANICE stems from a very different behavior of the Antarctic Ice Sheet component". Please could the authors provide the meaning that they refer to and explain what behavior specifically in Antarctica is different between the two ice models and why this has the effect in Indonesia. The final sentence of this paragraph is wholly unsatisfactory at explaining the difference in behavior. This is an example of the paper failing to offer insight beyond their study area and specific topic.

(6) Assumptions and approaches to reconstructing sea level

I have several queries about the specifics of how relative sea level was reconstructed.

(a) In the methods section, I would like to see plots of the water logger data and the correlations to one another and the tide gauge in Makassar.

(b) What is the tidal range at these sites? This key piece of information is missing.

(c) In figure 4, there are very large differences between sites for the height of living corals. Presumably, this is caused by similarly large differences in tidal range, but since tidal ranges for the sites are not presented anywhere in the paper it is impossible to confirm. Please could the authors provide this information and a supporting explanation as to why tidal range varies by so much over distances of less than about 100 km (Figure 1D; there is nothing in the figures to indicate that tidal range should vary dramatically among the sites). Alternatively, tidal range doesn't vary much between sites, in which case the authors need to explain why modern corals have different relationships to tides over the same small distances.

Figure 4 must also show the height of other important datums (particularly MLW, MLLW,

and LAT) that are used as part of the indicative range for dated corals. There must be explicit statements if these datums have the same or different relationships to mean tide level (and one another) at each site. The reader needs to see the data that supports the authors assertion that coral microatolls live between MLW and LAT (Line 84).

(d) Tables 1 and 2 must also show the original radiocarbon results (radiocarbon age, error, lab ID, d13C etc) because calibration curves and marine reservoir corrections change, but the original results will not. Adding this information will make the paper more useful in the long term.

(e) The equation used to reconstruct relative sea level (section 3, page 6) includes a term ("Er") for the amount of material eroded from the top of a coral microatoll. The authors assume that all coral microatolls had a pre-erosion thickness of $0.48 \pm 0.19$ m (line 175) based on a survey of modern corals in Mann et al. (2016). This assumption seems tenuous because the structure of a microatoll depends on the pattern of relative sea level change. For example, if sea level is rising then microatolls grow vertically (presumably getting thicker rather than wider). In contrast , if sea level is stable they grow laterally (presumably getting wider, but not thicker). Can the authors justify why the thickness of modern microatolls surveyed when sea level is rising rapidly (line 163) would the same as fossil microatolls that lived when sea level was rising more slowly, stable of falling across the mid-Holocene highstand?

The value of term Er should be presented in Table 2.

(7) Minor Points In figure 3, time should run from left to right.

Line 72: My understanding is that the eustatic sea level curves in ice models such as ICE 5G show the fastest rates of melt/sea-level rise during the Holocene occurred earlier than 6-3 ka BP.

Line 91: Why are index points from the Maldives, India and Sri Lanka mentioned here? This data is not used anywhere in this paper.

The structure of the paper would be improved by presenting modern coral surveys before the fossil samples (in methods, results, discussion and conclusions).

---

## Referee Comment (RC2) · Anonymous Referee #2 · 20 Aug 2019

This paper presents interesting new mid-late Holocene coral microatoll data from the Makassar Strait and produces a new regional RSL curve which is compared to GIA models. Although there is a good quantity of new data and the presentation of this data is generally good the broader implications for understanding past and future eustasy, or regional RSL are not well made. Each section of the discussion either falls short of making important new insights or reiterates statements from other papers (e.g. Mann et al, in review). As it stands this paper does not provide convincing reasons for the study taking place other than to document local RSL during the mid-late Holocene. Discussion – section 5.1. Absence of evidence for high RSL (over 1 m above MSL) does not categorically rule out the fact that RSL could have been higher in the Holocene, particularly at the start of the high stand when you have fewer index points. Your data

is not continuous, and in all cases has clustered SL index points from individual islands with millennial-scale time gaps. It is highly unlikely that the earlier (De Clerk/Tija) data is in situ but you cannot categorically rule out a slightly higher high stand earlier in the Holocene. You should therefore be slightly more cautious in describing your data. You also state that the re-analysis of the earlier work was largely undertaken in Mann et al. (in review) and therefore this discussion surely just repeats this analysis? How is section 5.1 in this paper different to what is discussed in Mann et al., in review? As that paper is in review I am not able to look at it for information. Discussion – section 5.2. Can you be clearer about what the Antarctic fluctuation during the Holocene is, that is causing the ANICE model to better fit your data? Does this model fit better elsewhere in this region? What larger implications of the model-data fit can you make using this dataset? I feel as if the broader significance of this section is not well explained, but this data-model fit may not be region-wide, so it may have no significance at all. If it is not this raises more serious questions about your data. Discussion – section 5.4 Is there a chance that the elevation data is incorrect for Barrang Lombo rather than there being subsidence on this island? If you are arguing that subsidence of $\sim$0.8 m has oc-curred since the mid-Holocene on this island only, is there any other geomorphological evidence of subsidence (lower elevation reef flat compared to other islands, or tilting of the reef flat surface?). Surely if water extraction and buildings are causing this subsi-dence it should be ongoing and therefore should be seen in the surface morphology of modern microatolls (or is the rate too small)? You should make more comment about this as a theory. Why is it significant that the modern microatolls are 'a few hundred meters away from the island'? Where were the fossil microatolls that were sampled in relation to the island? Are you arguing that the modern microatolls are not affected by subsidence because they are located further from the centre of the island? If you are indeed suggesting subsidence you need to substantiate this with other evidence beyond that derived from the microatoll data. Discussion – section 5.5 I don't think it is sensible to compare single-dated microatolls to the Common Era SL curve from Kopp et al (2016). Two SLIPs don't fit with the curve and there is no explanation for why this

might be. Why are these data not corrected for GIA? Given the large error terms on the SLIPs and the lack of explanation for why some data fits and some does not, is this kind of data suitable for assessing Common Era RSL in this region? I'm not really clear where this discussion is going or its value to the manuscript. Tables 2 and 3 – be clear that Age and RSL uncertainties are +/- errors. As you plot the data from De Klerk (1982) and Tija (1972) on Fig 6 it would be helpful to include the data for these index points in a table, probably table 3. Also please include the raw 14C data and lab codes for all dated index points so that they can be recalibrated in future if/when new calibration curves are developed. Fig 3 – why have present to the L of these plots? I would like to see present on the R of these plots. It would make sense to me to have the X axis scale the same for each one, even though it will squeeze the datapoints on some of the graphs. Fig 6 – marine limiting data points would normally have the horizontal bar at the top of the vertical distribution, not at the base. Your terrestrial limiting data point should be drawn with the horizontal line at the base of the vertical distribution. It would help if these data were included in table 3 so it is clear how you have plotted them. Why is tidal range not stated for each island location? This would help in interpreting the modern HLC data. I am not clear why you have used a constant erosion variable where erosion has occurred, as this is likely to vary over time.

Minor typographic corrections: Line 72 – of 'the' Holocene, insert extra 'the'. Line 80 – can span from MLW 'to' LAT not 'and'. Line 81 – 'reach' not 'reached'. Line 154 – 'second' not 'seconds'. Line 244 – is the elevation in m MSL? This isn't clear in the text. Line 308 – 'for which concerns' – rephrase, perhaps to 'Concerning. . .'. Line 373 – 'sites' not 'sited'. Line 430 – 'few main conclusions that concern' rather than 'few main conclusions for which concerns'.

---

## Editor Comment (EC1) · Pierre Francus (Editor) · 20 Aug 2019

Dear authors,

We have now received two reviewers reports.

As you have seen, their concerns are rather substantial. You are now supposed to prepare a response to these comments. Next, I will take your response into account to decide if I invite you or not to submit a revised manuscript. I would like to let you know that, based on the reviewers comments only, I do not expect to be encouraging you to submit a revised version.

Nevertheless, I'm looking forward to reading your response to both reviewers comments.

[Figure]

Kind regards

Pierre Francus
* * *

---

## Author Comment (AC1) · 16 Sep 2019

Dear Editor, We would like to thank you for the opportunity to answer the Reviewers comments and suggestions. While we acknowledge that both Reviewers are critical on several aspects of our paper, we also remark that Reviewer 2 made constructive suggestions and offered helpful ideas and recommendations. Reviewer 1 recommends that "a wholesale re-evaluation of the MS and underlying data are necessary". We think that re-writing some of the sections in light of the criticism raised by the two Reviewers and adding some accessory data and models will produce a stronger MS, that we hope will be acceptable in Climate of the Past. Hereafter, we try to summarize all concerns and ideas of both Reviewers, and we provide detailed comments to each of them. We hope you will give us the opportunity to revise the paper accordingly, as

we still believe that this could be a good contribution to Climate of the Past. If you choose to move forward asking us a revised version, we will send you the new MS with a detailed answer to each point raised by the Reviewers, including lines in the MS where the critical points are answered. Comments about how data are used in the paper Both Reviewers, in their opening statements, comment that this is a paper based on data. While Reviewer 2 comments the data characterizing the data as "interesting", "good quantity" and agrees that the presentation of data is "generally good", Reviewer 1 seems more negative about the fact that we present new data ("unfortunately, this paper is largely a description of some new data"). Both Reviewers agree in asking to explain better the rationale of this work, in order to give our data a larger context. In this paper, we present new data that was measured and dated as accurately as possible in an area of difficult access. A recent special issue in Quaternary Science Reviews reviewed hundreds of papers that reported, in total, more than 5500 data points. Each paper reported data exactly as we do in this MS. All together, these data are essential to validate GIA models at global scale. And validating GIA models in turn is important as GIA corrections are used, among other applications, to correct tide gauge data. We propose to restructure our introduction and rationale for the study to clarify this relationship between paleo sea level studies and current sea level, in order not to give the idea that our work is relevant only within the regional context. We can then take this point back in the conclusions to show what still needs to be done in SE Asia to reach a reasonable knowledge on Late Holocene RSL. In their comments, both Reviewers are asking for our raw data, specifically radiocarbon and tidal data. We commit to add all the original BETA Analytics reports on our radiocarbon ages and the water level data we collected in Indonesia. Tide Gauge data from Makassar tide gauge are private communication from BIG Indonesia, the National Geospatial Agency. The data have to be individually requested as per their rules, so we cannot put them as open access (The address to request the data is: Pusat Jaring Kontrol Geodesi dan Geodinamika, Bidang Jaring, Kontrol Gaya Berat dan Pasang Surut, Jl. Raya Jakarta-Bogor KM. 46, Cibinong 16911, Indonesia). We would like to remark, though, that these data were

used only to refer our measurements to MSL and not to interpret the paleo RSL as Reviewer 1 seems to point out.

Rationale of the work Reviewer 1 asks to address several questions within the introduction. The first is "Why it is necessary to document the height and timing of a Holocene sea level highstand given that it has been done so in many places already?" We agree that we should have made this more explicit. The brief answer is that, as sea level is spatially variable due to a number of factors, studying paleo sea level changes at different geographic locations is very relevant to understand patterns and timing of land/sea level changes. The presence and magnitude of the Holocene highstand in tropical areas is the result of the combined effects of eustatic history and glacio-hydro-isostatic adjustment (GIA). Documenting the Holocene highstand (its timing and elevation) at different places serves to better constrain how GIA and eustatic forcings are intertwined in both space and time. Our study provides yet another constraint on these processes. We plan to expand our introduction to explain this point better. Then, Reviewer 1 proceeds to ask: "What is the significance of SE Sulawesi? What can be learned by reconstructing relative sea level here that wasn't already known from existing studies?" We would like to remark that SE Sulawesi is not the area of interest of our study. Our Study area is located in SW Sulawesi. In general, southern Sulawesi is in the central part of Indonesia and thus is supposed to be a good region to study sea-level variability due to its central position within Indonesia. Furthermore, the study area we addressed is often reported as tectonically stable, while further north tsunamis and earthquakes are affecting the coast and hence there might be departures from eustasy due to tectonic activity. We propose to insert considerations on these matters in the Introduction, expanding on the relevance of our study area. Another matter raised by Reviewer 1 is that "Future sea-level rise is a convenient angle for making the topic of a paper appear widely relevant, but unless the paper leverages paleo results to produce better predictions this motivation should be removed". This matches a comment by Reviewer 2 asking to improve the description of the broader significance of our data. While our opening statement is about the importance of future sea level rise, we would like to

remark that we do not really frame our study around this topic. As briefly explained above, understanding late Holocene local sea level histories is indeed necessary to better analyze modern datasets. We can surely re-phrase the first paragraph of the introduction to better reflect the relevance of our study and its rationale. In the discussion, we propose to insert a new heading, "Implications on future sea level changes" to better explain what can be learned from our data in terms of future sea level rise.

Comments on Results section Other than asking to disclose our water logger data (which we will do, see above), Reviewer 1 asks to modify our Figure 4 to show also sea level datums together with the height of living microatolls. This is, according to the Reviewer, necessary as they are "used as part of the indicative range for dated corals". We think that there is a bit of confusion here, and we apologize if this stems from our choice of wording. We will revise our methods section carefully to make sure that it is clear that we did not use the tidal ranges as part of the indicative meaning of our microatolls. We think that this misunderstanding may originate from the fact that, in the introduction, we summarize how microatolls are generally interpreted: using tidal datums. This is well established in literature, and we do not think we need to show (as Reviewer 1 asks) that this is true, as we provide several references for it. In our section 5.3 we discuss that using the height of living corals is a better option than using tidal datums because it allows to take into account small regional differences, such as the ones we propose for the strait of Makassar. Unfortunately, in this area there is only one tide gauge, in the city of Makassar, and the few days of water logging we have, do not allow to establish tidal datums rigorously. As Reviewer 2 also asks to plot tidal ranges in our MS, we can propose to enrich our results with a tidal model forced with offshore constraints and calculate tidal datums at our locations and see how they match with the height of living corals. As this would take a bit of additional work on our side (that we have the capabilities to do, but tidal models ) and would require to model a few different sea level scenarios (e.g. higher sea levels during the highstand) we would like to ask the Editor if he thinks that adding this part is necessary. Another comment related to our data by Reviewer 1 is that we estimated the erosion thickness of some of our microatolls. Reviewer 2 also comments on this, saying that erosion rates might vary with time. We clarify that not all microatolls have been corrected, we will mark in the paper the ones that were (n=10). We also propose to mark them in the figures, so it will be immediately clear which microatolls might carry additional vertical uncertainties. We surely agree that using a single value (that was measured in the field by Mann et al., 2016) is a crude approach, but this is the only way we can take into account the erosion, that surely happened for some microatolls based on their morphology. Reviewer 1 suggests some interesting lines of discussion, that we propose to implement as caveats in our revised MS, expanding on them in the section where we discuss the erosion correction. Discussion – Abandoning conflicting sea level histories Both Reviewers express doubts about this section, also in light of the fact that Mann et al was in press at the time of the review. We shared a confidential copy with the Editor at the time of submission, and we are confident that the overlap with this MS is kept at a minimum. The paper is now published with this DOI https://doi.org/10.1016/j.quascirev.2019.07.007 and it is Open Access. This section is based on the comparison with older data in the same region, something that has been discussed previously by Mann et al., 2016 and Mann et al., 2019. In both papers, we have been cautious to reject these data, as we are well aware of the implications of doing this (i.e., eliminating a possibly high Holocene highstand from the Makassar Strait). Mann et al., 2019 write: Following the discussion about possible sources for RSL data inconsistencies in the SEAMIS database, site-specific discrepancies between [. . .] Tjia et al. (1972) (sub-region #5b) and de Klerk (1982) and Mann et al. (2016) (sub-region #6) must be resolved with additional high-accuracy RSL data before the existing datasets can be used to decipher regional driving processes of Holocene RSL change within SE Asia. Mann et al., 2016 already proposed that these data may represent storm deposits. Here we expand on this point showing wave heights in the region, opening also up to the fact that, despite the Spermonde Archipelago shelf is considered as tectonically stable, historical earthquakes have been recorded further north and waves may have propagated into the shelf. We checked if there is the possibility to model at least how one event would propagate

onto the shelf, but unfortunately, we miss realistic earthquake parameters to run our model. Again, we could propose to use a simple hydrodynamic model, forced with one historical storm, to show whether these high deposits can be explained. As we have no precise topographic data from the Tija and De Klerk studies we would have, though, to estimate a typical cross-shore profile (including bathymetry) and this might raise more questions than answers. Also on this matter, we defer to the Editor before attempting this modeling approach. Overall, we propose to expand the discussion in this section, also taking into account some comments on it by Reviewer 2 and making clear what is postulated by previous works and what is original here. Bottom-line, we feel that we have enough data from different islands to reject that the highstand was as high as 5-6 meters in this area. It is true that we are missing the highest peak (probably), but there it is very difficult to reconcile our data with a 5-meter highstand. Reviewer 2 seems to agree with this statement.

Discussion – Validation of GIA models Both Reviewers express their criticism on this section, asking in substance to expand our considerations on the underlying ice models. Reviewer 1 asks specifically "should we discard ICE5g because its fit to data in South East Sulawesi is worse than ANICE?" We would like to remark that nowhere in the paper we give the idea that one model should be discarded over another. We just note that, in our study area, one model matches data better than another does. This is clear from line 439 in our paper where we state "some iterations of ANICE seem to perform better" and we go on arguing that more ice sheet and earth models should be made available to "compare with RSL data in search for a better match". Given the Reviewer's comments, we decided to take on our own advice, also in order to expand this section as the both Reviewers seem to welcome. We now have 54 different ice-earth model iterations compared to the 8 we presented in the paper. We ran not only ANICE, but also ICE5G and ICE6G iterations with varying mantle viscosities. We share hereafter some preliminary plots to show the Reviewers that we are now in a better position to comment on the GIA points they raise. First of all, the figure below (Fig 1) shows the results of different earth viscosities associated with ANICE (red), ICE5g (blue) and

ICE6g (green). ANICE is still the one giving a lower highstand (left panel) and this is due to the ice history before the highstand itself (right panel). Using these new model runs, we first plan to use neighbouring areas from Mann et al., 2019 to gauge whether sea level indicators dating 10-12ka match better ANICE or ICE5g – ICE6g melting patters. Then, we will move on to compare our data to the model results. In general, we will maintain the notion that one single area cannot be used to say that ANICE performs better overall (we know well that it takes much more to choose a model over another for a given region). But it appears that our data (with the exclusion of Barrang Lompo, for which we discuss subsidence) fit better models with a low highstand (see below). Having these new models available is interesting, and grants some further discussion that we propose to add to the GIA section. For example, taking a snapshot at 5ka of the three models highlighted in the figure above (fig. 2) with full colors, we see that they show very different RSL histories in SE Asia, with ICE5G and ICE6G being essentially very similar and ANICE producing overall a very small highstand (see images below, respectively ANICE – ICE5G – ICE6G compared at 5ka and one mantle viscosity). We think that these new model outputs can be included in the paper, and producing a set of maps at regional scale such as those shown above (and at different times) (fig. 3) should help clarify. Overall, we propose to restructure the discussion of the GIA session giving a set of model maps that can be used by other workers to test whether the best fitting models in our area is compatible with other areas. Using tectonically stable areas in our database, we might also attempt to add to these maps points indicating when and how high was the highstand, taken from an update of Mann et al., 2019 (few data have been published since then).

Discussion – Local subsidence effects Both Reviewers are skeptical to the part of the discussion where we point to the fact that one heavily populated island might be subsiding due to local groundwater extraction and the weight of buildings. On this respect, Reviewer 1 seems more skeptical ("this explanation seems unfeasible to me"), while Reviewer 2 is more prone to consider it as a theory ("you should make more comments about this as a theory"). We remark that we left this as a hypothesis, as in the

discussion and in our conclusive point we use the conditional. In the revised version, we will advise further caution to interpret this result. We are currently trying to see if there is anything we can do to provide additional context, as suggested by Reviewer 2. One possibility we are exploring is to look at InSAR data with the help of a collaborator. We propose to report on this effort in the new version of the paper. We also plan to discuss further on the rates, comparing the subsidence of $0.8\pm0.3$ cm/year to other local subsidence rates in similar contexts. We are afraid, though, that there will not be many examples due to the lack of precise long-term surveying at small islands such as Barrang Lompo. Reviewer 1 is also questioning that Barrang Lompo data are really lower than the other areas, invoking "clustering" of our data. Reviewer 2 asks if it is possible that our data are wrong in Barrang Lompo. We would tend to exclude this latter possibility, as the survey methods we adopted are solid (levelling is among the most reliable surveying methods available). For which concerns data "clustering", we present hereafter a magnified version of our data (fig. 4) that might be the basis for further discussion. It appears obvious, to us, that the main clustering is the one we highlight in the paper, i.e. Barrang Lompo versus all the other islands. There is a second, minor discrepancy starting around 4700 BP between Panambungan/Bone Batang and Sanrobengi. This might be worth discussing, as these two islands are located in a different geographic setting, with Sanrobengi closer to shore (see Google Earth image below, fig 5.). Keeping in mind that these differences are, at best, 20 cm if we take into account error bars, it might be possible that these islands might have been subject to slightly different isostatic histories due to, for example, sediment loading or water loading of the shelf. The graph and brief discussion above also answers Reviewer 1 commenting that "new sea level histories to be conflicting within the study area". While Reviewer 1 comments that Bone Batang and Sanrobengi appear at odds, we remark that they seem perfectly overlapping within time and vertical error bars. We understand, though, that this comment originates from the way we chose to plot the data. We propose to insert, in the final version of the MS, a graph similar to the one above and discuss the inter-site discrepancies between entering the discussion of the

Barrang Lompo data.

Discussion – Common Era Both Reviewers point out that our discussion of the Common Era is not well constrained, and it appears not well related to the rest of the MS. Reviewer 1 is correct in saying that we did not investigate the existing data properly, also in light that, when we wrote this paper, we had the Mann et al. database in our hands. In the latest update (that contains data that were not published before the Mann et al., 2019 paper was out) we found out that there are 200 data points dating between 0 and 3000 BP. The co-authors are currently debating whether to delete completely or expand, as requested, the Common Era part of the discussion. Regardless of the outcome of the discussion, we will obviously re-write our discussion and focus it on SE Asia, explaining better the meaning of our data in the regional context. Minor points We plan to carefully consider and wherever possible accept the minor points raised by the Reviewers.

Please also note the supplement to this comment:
https://www.clim-past-discuss.net/cp-2019-63/cp-2019-63-AC1-supplement.pdf
* * *
[Figure]

**Fig. 1.** Results of different earth viscosities. ANICE (red), ICE5g (blue, ICE6g (green). Right panel: ice history before the highstand and left panel: ANICE giving a low highstand.

[Figure]

**Fig. 2.** The data fit better with a low highstand

**Fig. 3.** Maps of different model outputs. Please find further explanation in the text

[Figure]

**Fig. 4.** Close up of our data. Barrang Lompo provides deeper positions than the other islands

[Figure]

**Fig. 5.** Google maps picture showing the locations of Sanrobengi and Bone Batang

---

## Author Comment (AC2) · 23 Sep 2019

we addressed your comments in our AC1 to the editor (EC1)

———————————————

---

## Author Comment (AC3) · 23 Sep 2019

we addressed your comments in our AC1 to the editor

---

## Author Comment (AC4) · 23 Sep 2019

we addressed your comments in our AC1 to the editor

———————————————

---

## Author Response (AR1)

Object: Revisions of the Manuscript "*Holocene and Common Era sea level changes in the Makassar Strait, Indonesia*" by Maren Bender et al.

*27/12/2019*

Dear Editor,

We would like to thank you for giving us the possibility to revise the MS in object. After careful consideration of the comments of two anonymous reviewers, we now resubmit a substantially reworked version of our previous manuscript. While our core results remain the same (i.e., 24 fossil microatolls dated in the Spermonde Archipelago, Indonesia), in comparison to the previous version we now extended the number of GIA models to which we compare our relative sea level observations. We believe that this gives us the possibility to discuss our data more in detail. Overall, we would like to draw your attention on six main points:

1) The MS has been largely rewritten, taking into account the reviewers suggestions and trying to better describe our data and their meaning.
2) In conjunction with the MS, we present a set of Supplementary Materials containing not only the data we surveyed and the models we used, but also the scripts we employed to make the key figures of our study. We hope this will be useful for other scientists who will work with datasets similar to ours.
3) We provide our GIA models for a large region encompassing South and Southeast Asia. We believe that this will be of interest for scientists working on Holocene sea levels in this region: to our knowledge, the only other models readily available for this area are the five we recently published here: https://www.sciencedirect.com/science/article/pii/S2352340919309552#tbl1. This new dataset enlarges the previous one tenfold (54 models iterations in total).
4) Besides the GIA models, we share everything we measured (e.g., water levels, elevations, radiocarbon ages) in the Spermonde Archipelago.
5) One colleague, Julia Illigner, helped us with a better definition of the survey errors and on the analysis of water level data against the Makassar tide gauge. For this reason, she was added to the co-authors.
6) The new MS contains four tables (one more than the previous version) and 12 figures (four more than the previous version). Overall, we revised almost all our figures to improve clarity. The new figures were produced to better illustrate our points and answer to the reviewer comments.

We would like to thank the constructive criticism by the two reviewers: their suggestions pushed us to be more critical towards our data and our approaches. We believe that this resulted in a much stronger paper and in a wealth of open-access data (e.g., our GIA models and water levels) that will surely benefit the sea-level community working in the broader S and SE Asia region.

We are confident that this round of reviews produced a better MS than the one originally submitted. Please find hereafter detailed answers to the reviewers, pointing to where, in the MS, we took their suggestions into account. Reviewer's text is highlighted in gray, our answers are in plain text below. Central reviewer comments and our specific replies are in bold.

Best Regards, Maren Bender and the MS co-authors.

**Reviewer 1 (R1)**

This paper develops 24 new sea-level index points using radiocarbon-dated coral microatolls from South East Sulawesi in Indonesia. The dates span the second half of the Holocene (past 6 ka) and capture a time period where Earth-ice models predict a highstand in sea level. To be suitable for publication in Climate of the Past (or elsewhere) I believe that **a wholesale re-evaluation of the manuscript and underlying data are necessary**.

We thank R1 for the time they dedicated to the revision of the MS. As the reviewer suggested, **we did a wholesale re-evaluation of the MS**. We rewrote several sections, also considering new data and new lines of discussion. We added new figures and revised the previous ones for clarity. While we revised slightly the error bars on our data, the underlying data is the same as the previous version and we stand by them. Nevertheless, we added 49 new GIA models to the previous 5 runs. This allowed us to better discuss our RSL data in light of other processes. All our data and models are attached to the MS in the form of Supplementary Materials. We make also available the scripts we used to make the main plots of our MS.

(1) What is the purpose of this study? **Unfortunately, this paper is largely a description of some new data**. The reader is not provided with a specific and compelling reason for why the work was undertaken in the first place, or why the study sites and region are important. Furthermore, there is no explicit take home message about the wider implications of the results. As such, the current manuscript is only of local interest to others studying coral microatolls and sea level near Makassar, Indonesia. In the introduction the authors should provide an explicit motivation for their work. Please explain **(1) why is it necessary to document the height and timing of a Holocene sea level highstand** given that it has been done in so many other places already? (2) What is the significance of SE Sulawesi? What can be learned by reconstructing relative sea level here that wasn't already known from existing studies? The abstract (lines 38-40) and introduction (lines 55-61) frame this study around the threat of future sealevel rise. **However, the paper makes no effort to use the results in this context (see conclusions for example).** Future sea-level rise is a convenient angle for making the topic of a paper appear widely relevant, but unless the paper leverages paleo results to produce better predictions this motivation should be removed.

To answer this comment, we would like to remark that both this version and the previous one contain **not only RSL data, but also GIA models at a very broad regional scale (beyond our study area)**. We hope that the restructuring of the MS gives now more insights on why we embarked in this study. Departures between observations and GIA models at Holocene time scales are often used to gauge long-term vertical crustal movements due to, e.g. tectonics or subsidence, that need to be taken into account when extrapolating future sea level rise scenarios. **This was the reason for the hype on future sea level studies, which was removed from the introduction in this version**. While we removed the part of the introduction explicitly mentioning future sea level rise, we remark that, in general, the main purpose of studying paleo sea level changes is to leverage better future sea level predictions at either global or local scale. Therefore, we maintain that studies, such as this one, reporting accurately measured and interpreted data coupled with state-of-the-art models are, indeed, widely relevant. To answer the criticism of the reviewer more directly, **in the Introduction (paragraph 2) we now explain why it is important to understand the Holocene high stand at different locations**. The new section "Paleo to

modern RSL changes" shows why the data and models we present in this paper **are important at a very broad geographic scale**.

(2) Mechanisms of Subsidence (section 5.4). One of the five sites (Barung Lompo) records relative sea level that is anomalously low (Figure 3E). The authors propose that the reason for this trend is subsidence and go onto to propose that loading by approximately 4500 people living there, building concrete docks, and extracting water from wells is the driver. **This explanation seems unfeasible to me. The amount of subsidence is large** (order 1m per figures 3 and 7 and line 383), must be very fast (**order of 1cm/year assuming that the development occurred in the last 100 years**), and it doesn't seem unusual to me that this level of development would occur on a coral island in which case surely there would be lots of islands in many tropical places showing rapid subsidence at locations all around the world. Furthermore, the subsidence appears to be restricted spatially to just the island itself and doesn't extend onto the adjacent reef flat (line 398). Rather than offering an unsubstantiated idea, please **could the authors provide some evidence in the form of (for example) loading calculations or discussion of instrumented examples from other inhabited coral islands that are subsiding**. Figure 7 is not necessary. Its very easy to see from figure 3 that the difference is about 1m. Why doesn't the peak in the smooth curve line up with the peak in the histogram?

This line of evidence is now discussed in the section entitled "Mismatch of the record in Barrang Lompo island". Here, as also suggested by R2, **we adopt a very cautious tone** for which concerns our interpretation of the mechanisms of local subsidence. The comment by R1 pushed us to do a bit more research on the population of Barrang Lompo. We found that the island is populated since at least 1720. This would translate into an **anthropogenic subsidence rate of 3-11 mm/yr** (depending on the assumptions on timing of population and uncertainties on the RSL observation differences). This subsidence rate is surely large, but it **is one order of magnitude smaller than anthropogenic-induced subsidence in coastal megacities in SE Asia** (up to tens cm/year). For which concerns investigating other lines of evidence, we tried to understand if InSAR data could be used to detect subsidence in Barrang Lompo, but unfortunately the island extent is too small to apply this technique. We did not find studies dealing with instrumental records of differential RSL changes at small inhabited coral islands, but the mechanism for subsidence proposed here is also reported by Church et al., 2006 (Global and Planetary Change), who state:

*"A thorough analysis of survey data at Funafuti (Kilonsky, personal communication) shows that the land adjacent to the UHSLC gauge is sinking relative to the land adjacent to the NTC tide-gauge benchmark (about 2.5 km away) by 0.6 mm yr⁻¹. This tilting may be caused by tectonic movement or (most probably) local subsidence (for example, due to groundwater withdrawal) and demonstrates that even on a single island, the relative sea-level trend may differ by as much as 0.6 mm yr⁻¹. In addition, the UHSLC gauge is sinking on its foundations by an additional 0.6 mm yr⁻¹, giving a total sinking rate for the UHSLC gauge of 1.2 mm yr⁻¹."*

This reference is now discussed in the paper. In partial support to our hypothesis, **we also report some observations of intense coastal erosion reported for Barrang Lompo**, which might be related to exacerbated relative sea level rise rates. We close the newly designed section on Barrang Lompo by **suggesting strategies to test our hypothesis**. We hope that our work on this part provides a sufficiently balanced view on the data and on our interpretations.

(3) Common Era sea level (section 5.5). The authors have 8 microatolls that formed in the last 400 years and these form the basis for this section of the discussion. I see no purpose for this discussion and was unable to see its relevance to the paper or wider discussions about Common Era sea level. The database used by Kopp (2016) is not an exhaustive list of all sea level index points, it states clearly that the data are limited to detailed records. **There are likely 100s of sea-level index points that were NOT included in Kopp**. The authors claim that their new sea-level index points are the first from South East Asia for the Common Era because there is no data in Kopp (2016) from this region. This is simply not true and ignorance of the literature is an inadequate caveat. As one simple and widely known example see Horton et al. (2005; https://journals.sagepub.com/doi/10.1191/0959683605hl891rp). I expect the World Atlas of Holocene Sea-Level Curves would probably also provide some examples of Common Era sea-level index points from South East Asia.

**Why is the Common Era even discussed?** What is the importance of this time period (as distinct from the late Holocene which this paper focuses on), particularly given that the highstand occurs earlier? Figure 8C shows that the new index points have uncertainties that are too large to be useful in inferring much about Common Era sea level.

The section to which this comment refers to is now entitled "Common Era microatolls". There was a long discussion, between the authors, on whether to toss this entire section. We decided to keep this in because, as it is true that there are several reports of Common Era sea level indicators in SE Asia, **but only one data point is reported for the time frame covered by our microatolls**. While we agree with the reviewer that the error bars are too large to gather anything significant on the eustatic sea level, we now use them to make a simple calculation of potential natural subsidence rates in our study area, which we then use in the following paragraphs. While we maintain that our calculations are only tentative, we believe that this section adds more context to our conclusions.

(4) Conflicting sea-level histories (section 5.1). This section of the paper (as written) seemed unnecessary to me. The authors discuss the Tija (1972) and De Klerk (1982) sea level reconstructions, but explain that a paper by Mann et al. (under review) contradicts the original interpretations of the two older papers. Why does this paper need to summarize the arguments made in Mann et al?

**To retain this section, the authors need to show the Tija (1972) and De Clerk (1982) data and provide a through explanation as to why those interpretations should be refuted.** There are major contradictions in this section. Line 316 states that "Makassar Strait being one of the most tsunamigenic regions in Indonesia". Line 330 states that this study and others before it assumed that Makassar Strait was tectonically stable. How can the authors justify these two statements – they seem fundamentally opposed to one another. Is tectonics on or off the table for interpreting the new relative sea level curves?

I think that **figure 3 shows the new sea level histories to be conflicting within the study area**. The authors should focus on this rather than Mann's (under review) of old data. Although Barrang Lompo is identified as an anomalous site, there are others that don't match up particularly well with one another. For example, Bone Bafang (Figure 3F) seems to sit systematically lower than Sanrobengi (Figure 3B) by about 25-30 cm. Why is this the case? Its hard to evaluate the significance of this difference without knowing tidal range (not provided anywhere in the paper). Why is the relative sea level fall at Panambungan (Figure 3G) at 6 to 5 ka BP not seen anywhere else? None of these conflicting sea-level histories are discussed (and much less explained) by the paper. These discrepancies seem very large

given how close the sites are to one another (within about 100 km) and until this variability is explained the authors cannot justify their later interpretations of regional trends in relation to GIA models or Common Era trends for example.

We decided to retain this section. **As suggested by the reviewer, we expand on the descriptions by Tjia and De Klerk** (both in the text and Table 4 and Figure 7). We expand on our arguments for rejecting these data points, and we provide possible explanations for these higher deposits. While we still mention tsunamis, we clarify that the most active **tsunamigenic region is much further North**. We also point out that the fact that a region might get a tsunami wave does not preclude its tectonic stability: the Maldives are overall tectonically stable, but they were hit by the 2004 boxing day tsunami. Yet, we are very cautious in mentioning tsunamis in this region.

For which concerns the potentially conflicting sea level data within our islands, we now provide a new layout for Figure 3, which shows more clearly the RSL data in the region. We think it is clear that all our microatolls, except those from Barrang Lompo, **provide a consistent sea level history within their error bars**. It is true that there our data might indicate some high-to-low swings of few decimeters, but some of them are indeed present in our RSL models. The overall consistency of our record is now also shown in the newly produced Figure 9, that indicated that **there are only two clusters in our dataset: that of Barrang Lompo and that from all other islands**. The observation of the reviewer based on the differences between single points should be avoided, as every SL index point should be considered within their standard deviation.

(5) Validation of GIA models (section 5.2) This section of the paper offers minimal insight. The new reconstructions are compared to some GIA models. Unsurprisingly, there is a spread of highstand predictions (height and timing) among the different GIA models and therefore the agreement to the new data also varies among models. The authors proceed to identify models that fit the data better and worse. **What should a reader take away from this section?** Should we discard ICE5G because its fit to data in South East Sulawesi is worse than ANICE?

**Why does the fit in this region offer more insight that the fit in other regions** (particularly those with much bigger and longer datasets such as the British Isles or South East Asia, where databases on index points have long been compared to GIA predictions). Line 344 states "The better match of ANICE to our data has a meaning for which concerns ice melting patterns. In fact, the lower highstand predicted from ANICE stems from a very different behavior of the Antarctic Ice Sheet component". Please could the authors provide the meaning that they refer to and explain what behavior specifically in Antarctica is different between the two ice models and why this has the effect in Indonesia. The final sentence of this paragraph is wholly unsatisfactory at explaining the difference in behavior. This is an example of the paper failing to offer insight beyond their study area and specific topic.

Taking into account the comment of Rev.1, we heavily edited this section, which is now titled "Comparison with GIA models". First of all, **we now provide a much larger set of models** that are compared to our data. To our knowledge, **this is one of the largest GIA model ensambles published for S and SE Asia, and we provide NetCDF files** that can be used to compare observation to modeled RSL in a broad region, together with the scripts needed to reproduce the figures we present in this paper also for other areas.

We also modified the text in order to **make it clear that we are not favoring one model over the other**. Also using some considerations on tectonics and subsidence from different hypotheses, we point out that different models fitting in our region have different meanings also for current sea level studies. As we moved away from the search for a "single fit", we also deleted the discussion related to the ice sheet patterns of ANICE vs ICE5g vs ICE6g that we agree were out-of-scope.

(6) Assumptions and approaches to reconstructing sea level I have several queries about the specifics of how relative sea level was reconstructed. (a) In the methods section, I would like to see plots of the water logger data and the correlations to one another and the tide gauge in Makassar.

The water level logger data are now reported in SM1. We cannot report the tide gauge data of Makassar as these data are, unfortunately, not in the public domain. Data can be requested to Badan Informasi Geospasial (BIG), Makassar. Unfortunately, we have no control over their public availability.

(b) What is the tidal range at these sites? This key piece of information is missing.

Unfortunately, we would need 19 years of water level data to gather tidal ranges at the sites studied. This is why the information is missing in the paper. Nevertheless, in the "Regional Setting" section, we inserted what we can derive from the Makassar Tide gauge: "*In the Spermonde Archipelago, the tidal cycle is mixed semi-diurnal with a maximum tidal range of 1.5 m (data from Badan Informasi Geospasial, Cibinong/Indonesia).*". This is the same information contained in Mann et al., 2016.

(c) In figure 4, there are very large differences between sites for the height of living corals. Presumably, this is caused by similarly large differences in tidal range, **but since tidal ranges for the sites are not presented anywhere in the paper it is impossible to confirm**. Please could the authors provide this information and a supporting explanation as to why tidal range varies by so much over distances of less than about 100 km (Figure 1D; there is nothing in the figures to indicate that tidal range should vary dramatically among the sites). Alternatively, tidal range doesn't vary much between sites, in which case the authors need to explain why modern corals have different relationships to tides over the same small distances.

Unfortunately, as explained above, **we cannot present the tidal ranges for the sites as this would require very long-term water level measurements**. What we do is to point out the difference we find in the living microatolls, and advise against using a single tidal range to derive the indicative range also as such small spatial scale. The comment of the reviewer, though, pushed us to investigate more this onshore-offshore gradient, and we found out that a similar pattern is **discernible in our water level data**. We plotted Figure 4b to show it, but caution must be adopted in comparing directly Figure 4a and 4b: the water levels of 4a represent, possibly, MLLW/LAT ranges over 50-100 years (living microatoll time), while 4b presents only a snapshot of a couple of days.

Figure 4 must also show the height of other important datums (particularly MLW, MLLW, and LAT) that **are used as part of the indicative range for dated corals**. There must be explicit statements if these datums have the same or different relationships to mean tide level (and one another) at each site. The reader needs to see the data that supports the authors assertion that coral microatolls live between MLW and LAT (Line 84).

We remark that **we do not use tidal ranges to calculate the indicative range for dated corals**. In the new version of the paper, we tried to state this more clearly to avoid this confusion. The reason why we

used HLC to calculate the indicative range is, in fact, that **tidal datums are not available for each site within our study area**. We only have few days of water level data, which is now attached as SM1.

(d) Tables 1 and 2 must also show the original radiocarbon results (radiocarbon age, error, lab ID, d13C etc) because calibration curves and marine reservoir corrections change, but the original results will not. Adding this information will make the paper more useful in the long term.

We totally agree. In SM3, we share all the data received from the radiocarbon analysis laboratory.

(e) The equation used to reconstruct relative sea level (section 3, page 6) includes a term ("Er") for the amount of material eroded from the top of a coral microatoll. The authors assume that all coral microatolls had a pre-erosion thickness of 0.48 ± 0.19 m (line 175) based on a survey of modern corals in Mann et al. (2016). **This assumption seems tenuous** because the structure of a microatoll depends on the pattern of relative sea level change. For example, if sea level is rising then microatolls grow vertically (presumably getting thicker rather than wider). In contrast , if sea level is stable they grow laterally (presumably getting wider, but not thicker). Can the authors justify why the thickness of modern microatolls surveyed when sea level is rising rapidly (line 163) would the same as fossil microatolls that lived when sea level was rising more slowly, stable of falling across the mid-Holocene highstand? The value of term Er should be presented in Table 2.

The value of Er is now reported in Table 2 and clearly marked in Figure 3 (grey error bars). **Modern microatolls thickness is the only proxy we have to account for how much material has been eroded**. **We put a caveat in our methods section** to reflect this very relevant consideration offered by the reviewer. Here is the sentence we inserted:

*We remark that this calculation does not take into account the fact that modern microatolls are thicker rather than wider because of the current rapidly rising sea level. In contrast, under Late Holocene falling or stable sea level changes, they were presumably getting wider, but not thicker. Hence, in our calculations, the added Er might be overestimated, as it is based on modern microatoll proxies.*
* * *
**Reviewer 2 (R2)**

This paper presents interesting new mid-late Holocene coral microatoll data from the Makassar Strait and produces a new regional RSL curve which is compared to GIA models. Although there is a good quantity of new data and the presentation of this data is generally good **the broader implications for understanding past and future eustasy, or regional RSL are not well made**. Each section of the discussion either falls short of making important new insights or reiterates statements from other papers (e.g. Mann et al, in review). As it stands this paper does not provide convincing reasons for the study taking place other than to document local RSL during the mid-late Holocene.

We would like to thank R2 for the time they dedicated to assess our MS. We re-wrote large sections of our MS, including the discussions and conclusions, in order to **make more evident what are the broader implications of our study**. We hope that this answers their comment. We also tried to be specifically clear on what was already published in Mann et al. 2016/2019 and this study. Overall, we present more data and more models that we surveyed following the works of Mann et al.

Discussion – section 5.1. Absence of evidence for high RSL (over 1 m above MSL) **does not categorically rule out the fact that RSL could have been higher in the Holocene**, particularly at the start of the high stand when you have fewer index points. Your data is not continuous, and in all cases has clustered SL index points from individual islands with millennial-scale time gaps. It is highly unlikely that the earlier (De Clerk/Tija) data is in situ but you cannot categorically rule out a slightly higher high stand earlier in the Holocene. You should therefore **be slightly more cautious in describing your data**. You also state that the re-analysis of the earlier work was largely undertaken in Mann et al. (in review) and therefore this discussion surely just repeats this analysis? **How is section 5.1 in this paper different to what is discussed in Mann et al., in review?** As that paper is in review I am not able to look at it for information.

We tried to be more cautious in describing our data. In our results section we only provide an overview of the data and model results we have, with no interpretations. We point out, though, that we now have a very widespread amount of data compared to Mann et al., 2016 (who had two islands only), and the bulk of data points to the same conclusion: we find no evidence of higher RSL than our microatolls in the Spermonde Archipelago. Our models indicate that the highstand might pre-date our RSL index points, but we find no evidence for it.

The paper Mann et al, in review, is now published (2019). In this paper, we quote exactly what Mann et al., 2019 concluded and we use this as the starting point for further discussion. Basically, **the data presented in this paper were not yet available when Mann et al. compiled their review**. These new data give us **more confidence in rejecting Tjia and De Klerk data** and in looking for alternative explanations.

Discussion – section 5.2. Can you be clearer about **what the Antarctic fluctuation during the Holocene is, that is causing the ANICE model to better fit your data**? Does this model fit better elsewhere in this region? What larger implications of the model-data fit can you make using this dataset? I feel as if the broader significance of this section is not well explained, but this data-model fit may not be region-wide, so it may have no significance at all. If it is not this raises more serious questions about your data.

Also following the comments of R1, **we now deleted this part**. We now show fits/misfits of our data with a larger set of GIA models under different tectonic uplift/stability/subsidence scenarios, and discuss what would change in terms of modern GIA rates. We believe that this gives a better frame to show how our data may be useful within a broader context.

Discussion – section 5.4 **Is there a chance that the elevation data is incorrect for Barrang Lombo** rather than there being subsidence on this island? If you are arguing that subsidence of 0.8 m has occurred since the mid-Holocene on this island only, is there any other geomorphological evidence of subsidence (lower elevation reef flat compared to other islands, or tilting of the reef flat surface?). Surely if water extraction and buildings are causing this subsidence it should be ongoing and therefore should be seen in the surface morphology of modern microatolls (or is the rate too small)? **You should make more comment about this as a theory.** Why is it significant that the modern microatolls are 'a few hundred meters away from the island'? Where were the fossil microatolls that were sampled in relation to the island? Are you arguing that the modern microatolls are not affected by subsidence because they are located further from the centre of the island? If you are indeed suggesting subsidence you need to substantiate this with other evidence beyond that derived from the microatoll data.

We carefully re-checked our data following this comment, and this led to some changes in how we propagated errors. While we believe that the new error bars are now more "solid", the overall dataset did not change much. Standing our measurement methods, **it is highly unlikely that there is an error in elevation data from Barrang Lompo**. Following this comment and that of R1 on the same section, we now made it more hypothetical, presenting the "human-induced" subsidence **more as a theory that needs testing**. One way to test it was suggested by R2: one might look at growth patterns of modern microatolls at different islands and see if there are differences. In re-writing the section in object, we tried to take into account all the questions raised by the reviewer on the previous version.

Discussion – section 5.5 **I don't think it is sensible to compare single-dated microatolls to the Common Era SL curve from Kopp et al (2016)**. Two SLIPs don't fit with the curve and there is no explanation for why this might be. Why are these data not corrected for GIA? Given the large error terms on the SLIPs and the lack of explanation for why some data fits and some does not, is this kind of data suitable for assessing Common Era RSL in this region? I'm not really clear where this discussion is going or its value to the manuscript.

**We deleted from the MS the comparison with Kopp et al.** Instead, we now use the microatoll data to gauge whether it is possible to calculate rates of natural subsidence, that are then used to discuss Late Holocene "stability" of the area. We refrain from any ESL consideration from these few scattered records, but we kept them in the paper as this particular time frame (the last 300-400 years) is almost not reported in SE Asia.

Tables 2 and 3 – be clear that Age and RSL uncertainties are +/- errors. As you plot the data from De Klerk (1982) and Tjia (1972) on Fig 6 it would be helpful to include the data for these index points in a table, probably table 3. Also please include the raw 14C data and lab codes for all dated index points so that they can be recalibrated in future if/when new calibration curves are developed.

We clarified that age and RSL uncertainties are +/- errors. We also included the data from De Klerk and Tjia in Table 4. The original lab analyses are available as SM3.

Fig 3 – why have present to the L of these plots? I would like to see present on the R of these plots. It would make sense to me to have the X axis scale the same for each one, even though it will squeeze the data points on some of the graphs.

Figure 3 was re-drawn to better show the data points. We feel that present to the Left or Right is a preference rather than a convention, and for the Holocene we have the feeling that many authors adopt the present to the left (see Khan et al., 2015 Quaternary Science Reviews, summarizing the Holocene sea level Atlas).

Fig 6 – marine limiting data points would normally have the horizontal bar at the top of the vertical distribution, not at the base. Your terrestrial limiting data point should be drawn with the horizontal line at the base of the vertical distribution. It would help if these data were included in table 3 so it is clear how you have plotted them.

We now have the same data plotted in Figure 3, and we changed marine and terrestrial limiting symbols to follow the convention highlighted by R2. All data plotted in figure 3 is now included in Table 2, 3, 4 and in the SM1 as spreadsheet.

Why is tidal range not stated for each island location? This would help in interpreting the modern HLC data. I am not clear why you have used a constant erosion variable where erosion has occurred, as this is likely to vary over time.

See answers to the same points raised by R1.

---

## Referee Report (RR1)

**Summary:** This manuscript contributes a reconstruction of paleo sea-level position at 24 points in time throughout the mid to late Holocene. Authors interpret the new sea-level index points from the age, elevation, and paleo-water depth interpretation of fossil fringing reefs across SE Sulawesi near Makassar, Indonesia. In contrast to other coral-based reconstructions of past sea-level changes, but in keeping with other Holocene reconstructions from the SW Pacific (e.g. Hallmann et al., 2018), authors evaluate fossil reef growth in a fascinating morphology – microatolls – highlighting the potential for reconstructing past sea-level in great detail using this approach. Additionally, authors combine new data with previous data from two other islands near Makassar (recalculated results of three previous studies to directly compare them), which demonstrate higher-than-present relative (local) sea-level (RSL) during the late Holocene.

Results and methods described here underscore the importance of evaluating fossil microatolls in the context of nearby living microatolls. Authors directly compare the elevations of fossil microatolls with their modern counterparts to evaluate the relationship between highest living coral (HLC) and the tidal cycle. Similar to previous works, this study finds that this relationship varies greatly on small spatial scales. To reconstruct paleo sea-level from fossil microatoll elevation requires the assumption that the relationship between HLC and the tidal cycle at that site has remained the same since the microatoll formed. Wisely, this study does not attempt to "connect the dots" and provide an RSL curve, but instead provides sea-level index points that can be compared with other Holocene RSL reconstructions and RSL predictions from GIA modeling. The results of this work contribute to scientific progress within the scope of Climate of the Past because they include reconstructions of past climate based on proxy data from a marine archive. Additionally, the data-model comparison may serve as motivation to identify GIA model parameters that accurately reconstruct Holocene sea-level data, and which may be applied to reconcile current sea-level trends and to improve model predictions of future sea-level changes. New paleo-observations of RSL provide critical constraints for models of future sea-level and ice-sheet behavior and the response of the solid earth to that behavior in a warming world.

I appreciate that the authors acknowledge that ages and elevations determined for fossil reefs – even microatolls with the potential for <1m vertical uncertainty – cannot be considered in isolation and directly translated into RSL. Instead, to constrain various sources of age and elevation uncertainty, post-depositional vertical land motion due to regional tectonism or GIA effects, local hydrography and oceanographic factors are considered prior to interpreting sea-level. Authors illustrate the importance of evaluating microatolls in the context of modern analogues to reference elevations to a tidal datum and place them in a precise sea-level reference frame. While they employ GIA modeling to consider the GIA signal in this region, they ultimately do not remove the GIA contribution from sea-level index points to estimate the glacio-eustatic contribution to sea-level. A strength of this manuscript is the transparent explanation of uncertainty calculations.

Overall, authors do not clearly define the main impetus for this work nor do they offer specific insight into future research directions, implications of results for the Holocene sea-level history beyond the Spermonde Archipelago. It is not clear until the conclusion that the purpose of the GIA model-data comparison is not to identify glacio-eustatic contributions to late Holocene sea-level nor to investigate the evolution of late Holocene sea-level but to identify a best-fitting GIA model that could be used to refine predictions of current and future sea-level trends. A best-fitting model is ultimately not identified, nor are future directions. I therefore recommend major revisions to this manuscript to improve clarity.

**Introduction:**
The introduction does not establish why it is important to reconstruct sea-level changes during the Holocene, a time of transition between glacial and interglacial climates, nor does it emphasize why SE Sulawesi was selected for analysis, the power of microatolls as a proxy for past sea-level position on multiple timescales, and why evaluating data in the context of GIA model reconstructions of past sea-level is critical to evaluating global mean sea-level from local sea-level reconstructions in the past and the present.

While the sea-level index points are combined with 3 previous studies from the vicinity of Makassar, there is no discussion of the results in the context of Late Holocene sea-level reconstructions from SE Asia and the South Pacific (e.g. Hallmann, 2018), or other Holocene SL reconstructions derived from microatolls. How do RSL results compare with other reconstructions and what are the limitations of the previous works? As written, it does not seem to be anchored to a clear history of previous work (regional, global, Holocene) for readers to critically evaluate the importance and significance of the results that it presents, nor is it framed as novel or distinct form previous work in terms of its methods, study site, etc other than the introduction of new index points and extensive GIA modeling.

I would restructure the first 75% of the introduction as follows:
1. Statement on importance of reconstructing Holocene ice-sheet and sea-level response to an interglacial climate. Succinctly state why this is relevant to accurately and precisely predicting timing and rates of future ice-sheet and sea-level response to the present warming climate. Why are you presenting new Late Holocene sea-level data and GIA models?
2. Briefly, describe state of knowledge from far-field, Indo-Pacific Holocene sea-level reconstructions – any trends or outstanding questions. Why are far-field records important? How you reconstruct sea-level index points using microatolls and what are their advantages/disadvantages relative to other sea-level indicators?
3. How do you expect GIA to influence local sea-level histories in this region and why it is necessary to correct local sea-level histories for the influence of GIA and vertical land motion due to tectonism in ordered to evaluate glacio-eustatic sea-level changes? As you already mention in the introduction, determining rates of subsidence/uplift due to regional tectonics by accurately estimating past sea-level is a circular problem discussed at length in Creveling et al. (2015). *
4. Why was SE Sulawesi selected for analysis and what steps did you take (as written) to generate an accurate RSL reconstruction?

Definitions: In the intro, define terms such as relative (local) sea-level (RSL) (Line 49), and what you mean by "eustatic" sea-level (Line 45 vs 50). On Line 45, you describe "globally averaged" sea-level, or GMSL (which is includes contributions from thermal expansion and changes in global ice-volume), but elsewhere (e.g. Line 50) you use "eustatic" to describe "glacio-eustatic" or "ice-equivalent" changes in sea-level in response to transfer of mass between ice and ocean (Mitrovica and Milne, 2002; Milne, 2015 *Handbook of Sea-level Research*, citations therein). It is my understanding that all three of the phenomena listed in Lines 51-54 to explain the common observation that far-field sea-level reconstructions record a mid-Holocene highstand fall under the definition of GIA (see concise explanation of equatorial syphoning/GIA trends and citations

in Dutton et al, 2015 *Science* in addition to Kopp et al., 2015, Mitrovica and Milne 2002, Milne and Mitrovica 2008, etc.). GIA processes include deformational, gravitational and rotational effects driven by the transfer of mass between ice and ocean that can cause local RSL changes to depart significantly from the GMSL curve or the response of the solid Earth and gravity field to the climate-driven surface ice- water mass redistribution (Milne and Shennan, 2013). Syphoning, changes in gravity due to surface ice-water mass redistribution and solid Earth deformation are all driven by GIA.

Minor points:
Line 44: I think that sections must be numbered. https://www.climate-of-the-past.net/for_authors/manuscript_preparation.html
Line 56: You repeat the definition of the RSL acronym again.
Line 65: To reconstruct paleo RSL, we **measured the age and elevation of microatolls,** ie…Line 71: fossil ones, that we **surveyed and** dated using radiocarbon.

**Methods**:
Lines 119 – 131: A conceptual figure or a reference to one may be useful here to visualize how microatolls are used as a proxy and linked to tidal datums, indicative range, etc. in this study.
Line 124: Please be more specific about what you mean by "extended periods of time" perhaps relative to the growth rate of the coral?
Line 130: Please be specific about what you mean by short-term sea-level fluctuations.
Line 140 – 141. This is an important point.
Line 141: Clarify your definition of indicative meaning.
Lines 117-141 General comment: Methods are clearly outlined. Assumptions made in using microatolls to reconstruct sea-level are not (e.g. as referenced in McLean et al., 1978). What assumptions go into assigning a reference water level to the coral's highest level of survival? Is the relationship between microatoll elevation and tidal cycle the same over time and across areas of the reef? Are all microatolls morphologically similar here and why is this a good field site?
Line 143: FMA's and LMA heights are the maximum (peak) height of the microatoll, correct? Or is it the average elevation surveyed across the top of the microatoll?
Line 167: Replace reducing with relating.
Line 177: How far away were these islands? Did you consider potential variations in the height of the geoid as per Woodroffe et al. (2012)?
Lines 204-222: Please explain further in the section on sampling and dating what kinds of samples you selected (slice of the microatoll? Hand samples?) and where you sampled from on the microatoll (the highest point on the microatoll or across it?). In general, how did you assign a radiocarbon age to a microatoll? Was there one date per microatoll or did you measure multiple dates to interpret an age (see distinction for U-series in Dutton et al., 2017)? Additionally, please clarify what diagenetic screening you employed when analyzing coral preservation in advance of radiocarbon dating and **report your XRD results (see more on XRD reporting in Vyverberg et al., 2018).** This information may also be useful in light of the documented erosion for most fossil microatolls in this study. Clarify your reference age for (a BP) – is the present defined as 1950 CE?

Line 223: Please clarify here or in methods why you are predicting RSL with GIA modeling. Later, in the discussion, perhaps touch on the following: *Do you intend to convert RSL to

GMSL via the extraction of the GIA signal at this location? Can any inferences of GMSL be made here? What steps would be needed to determine GMSL from your RSL results, and what are the challenges faced in converting RSL to GMSL via the extraction of the GIA signal at this location? Why don't you attempt to remove the GIA signal - provide clarification (e.g. further discussing implications of Fig. 11) as to why not. Elaborate on why evaluating GIA matters for Holocene/modern sea-level reconstructions and what the limitations to evaluating the GIA signal are here or in general.

**Results:**
Line 240: I see now that these are average radiocarbon ages as opposed to raw dates. I would clarify how many samples (dates) were analyzed to determine an age and how that data was evaluated for diagenetic alteration.
Line 178, 248, 551: Please revise phrasing of "For which concerns" to "concerning …"
Line 240: **Table 2:** Following the equation on line 172, RSL estimates in Table 2 appear to be off by ~ 0.01 – 0.02 m. (ex – PS_FMA1 Suranti: RSL = -1.46 – (-0.74) + 0.2 = -0.52 m. In Table 2 it is reported as -0.53 m and using the numbers in Sheet 9 of SM1 RSL = -0.54 m. The excel sheet reads -0.53m using whatever rounding rules were applied in excel and the data correction of +0.014m. Furthermore, the Reference Water Level reported in SM1 is not always comparable to that reported in Table 2. For Suranti and Tambakulu it is as -0.72 m in SM1 but reported as -0.74 in Table 2. Please address rounding and reporting discrepancies.

**Table 3:** Why is erosion error not included in Table 3, when it is included in SM1 Sheet 4 for FMA8 – 11 (Panambungan)? The erosion factor seems to be incorporated into RSL for those points in Table 3; without it the RSL values calculated in Table 3 are lower by 0.2m.

**Discussion:**
Line 318: Please clarify by how much HLC changes instead of "HLC changes substantially."
Line 345: I would mention what "sea-level data" refers to. It is only mentioned in the caption of Figure 7 that the earlier works used different sea-level proxies.
Line 349: Be careful about the use of "significantly" here and throughout. Is the difference statistically significant?
Line 358: I would elaborate on the differences between the proxies used in these different studies. How does the precision vary between them? Looking at the uncertainty bars in figure 3, the De Klerk and Tjia sea-level index points tend to be higher than those reported in Mann and this study, but they are also less precise. Several points from this study/Mann fall within the bounds of vertical error for points from De Klerk and Tjia.
Line 368: What additional data would need to be explored to evaluate the tectonics hypothesis?
Line 373: Are the De Klerk coral data collected from coral in growth position? (in situ)
Lines 519 – 560. It is to be expected that there is a range of highstand predictions that vary in space and time depending on GIA model ice and earth parameters, and it is clear that the fit between predicted and observed RSL also varies depending on the GIA model parameters as well as the assumed tectonic history. Is the main takeaway that the ICE5g model is not a good fit because, regardless of the tectonic history, the peak highstand predicted by ICE5g does not match the peak in the observed RSL data? What is the main outcome of this data-model comparison, or what steps can be taken to better compare them in future studies? The purpose of this comparison was not clearly defined in the first place, though the importance of identifying

GIA models that best fit Late Holocene data to improving model predictions of current and future sea-level changes is explained on line 553.
**How does the choice of GIA model affect the interpretation of RSL index points made in this study?** What can the reader conclude about late Holocene sea-level form the data-model comparison described in these latter sections and the earlier comparison of data between this and previous RSL reconstructions? See previous comments on line 223 regarding inferences of GMSL at this location. The discussion of Late Holocene RSL and Fig. 12 seemed to end abruptly. Please elaborate on why the results in Fig. 12 are widely relevant to modern sea-level estimates.
Line 572: Specify a gradient in elevation

**Figures:**
General comment: All figures have simple and elegant layouts. Fonts are legible and colors are clear. Sections are clearly marked and figures are overall helpful and easy to follow.
Figure 1: in 1b, I would not combine red and green-colored dots to make the figure accessible to color-blind readers. Perhaps try a dark boarder to yellow dots in c – i to make them more legible.
Figure 3a: Have you tried making the symbols slightly different between datapoints from this study and from Mann? As mentioned earlier do not include red/green together.
Figure 4a: I would identify the sites analyzed in this study with a (*) next to the name. Specify in the caption at least once that you mean individual microatolls instead of "individuals."
Figure 5: Change Red/Green combination.
Figure 11: Change Red/Green combination.  The four boxes in this figure are missing panel letters (a – d).
Figure 12: I would mark the position of the Spermonde Archipelago on the map for reference.

---

## Referee Report (RR2)

Department of Earth and Space Sciences | Merion Science Center, Room 207
West Chester University | West Chester, Pennsylvania 19383 | 610-436-2727 | fax: 610-436-3036 | www.wcupa.edu

February, 22, 2020

Dear editor and the authors,

I completed reviewing the manuscript "Late Holocene (0-6ka) sea-level changes in the Makassar Straight, Indonesia". The paper reports 24 new sea-level index points created to reconstruct paleo sea-level estimates during the last 6 ka. The index points are derived from 24 fossil microatolls from 5 islands of Spermonde Archipelago, Indonesia. The region is known as the far-field region, where the 6-3 ka sea-level highstand was suggested by previous researchers and predicted by GIA models. The higher than present sea level in equatorial region was explained by various mechanisms. The complexity of the processes resulted in special-temporal variability of sea level during Late Holocene and continues to impact the different areas along the coast at various rates. Further understanding of sea-level histories are essential for predictions of future sea level scenarios on local and regional scales. The manuscript reports new data along with previously published results and interpretation that address that problem.

High quality data presented by authors include age and elevation of fossil corals and their indicative meaning based on accurate calculations of high range of living microatolls. Authors made an afford to explain each of the applied uncertainty. The open-access data available as Supplement Materials also contained water level measurements, 54 GIA models with Jupyter notebook, and the scripts.  The new data was combined with 20 previously surveyed microatolls from the same archipelago and used for regional paleo relative sea level reconstruction.

Authors critically re-evaluated reported index points by De Klerk (1982) and Tjia et al. (1972) and suggested to reconsider sediment interpretation as high-magnitude storm deposits and until further field investigation exclude them from sea-level compilations.

I also carefully reviewed authors' responses to comments by two anonymous reviewers and concluded that the manuscript was significantly improved since its original submission and that authors critically addressed reviewers concerns and suggestions.

I suggest that the manuscript will be considered for publication after few minor revisions.

1. In the Abstract authors state that they are reporting 24 new index sea-level points (line 38). However, in the Conclusion the authors report 25 index points (line 556). It is my understanding, that microatoll PB-FMA 4 index point was rejected. Please clarify.
2. I suggest to add indexes "a" and "b" to the panels on Figure 8 to be consistent with other figures format.
3. I suggest to add indexes a, b, c, d to Figure 11. Text references to Figure 11 have already include the appropriate indexes (lines 530, 535, 539, and 542).

In addition, I agree with R1's comment 2 regarding the anthropogenic subsidence on Barrang Lompo island being the major reason for a low rate of sea level rise. Since the instrumental data to support the proposed hypothesis does not exist, authors suggest that high rate of coastal erosion on the island could be indirect evidence of human impact and propose to further investigate this idea or leave the question open inviting other plausible explanation of the low rate that mismatch the regional sea level trend.

In the summary, I believe that the manuscript presents valuable data and paleo sea-level reconstruction using best-fit GIS model and is suitable for publication in CP. Analysis of ice models beyond the study area empathizing the need for GIA correction as essential for estimate of eustatic sea-level changes and future predictions presents an interest to a broader scientific community.

Sincerely,

Daria Nikitina
Professor of Geomorphology
Department of Earth and Space Sciences
West Chester University

---

## Author Response (AR2)

Object: Revisions of the Manuscript "Late Holocene (0-6ka) sea-level changes in the Makassar Strait, Indonesia" by Maren Bender et al.

*13/03/2020*

Dear Editor,

Please find attached our reworked version of the manuscript and our detailed answers to the comments of reviewers 2, 3 and 4. While we changed some points as suggested by the reviewers, we also stand by our text and ideas in some instances. Our changes to the text and rebuttals are explained in detail below.

We would like to thank the suggestions and ideas to optimize the MS by the three reviewers that helped improving the MS.

The Reviewer's text is highlighted in gray, our answers are in plain text below.

Best Regards,

Maren Bender and the MS co-authors.

Line 89 – island not islands.

This was changed accordingly

Line 184 onwards – I am not totally happy with applying a modern-derived erosion value to these microatolls. What would you lose from your dataset if you removed them from your final analysis? The SLIPs with calculated erosion values (grey vertical error bars on fig 3) seem to sit systematically above the un-eroded ones. So is there a bias in your data caused by using this standard correction? Your final RSL curve would be more precise without them. I realise this would leave you with not that much to add to the Mann 2016 study, but the two sites you present with uneroded data match very well with the Mann data, which is a good thing. Another thing to consider would be to plot the eroded data completely differently (e.g. as boxes) to make it clearer that they are less certain. You also need to say something in the text about this apparent systematic offset between eroded (corrected) and non-eroded data.

We thank RV2 for this constructive comment. Of course, we could take these microatolls out of our study, but we believe that they present valuable RSL information. We believe that (also as a result of previous reviews) we labelled these microatolls clearly enough in figures, tables and in the text to present the reader an objective view on their reliability. In general, we note that the offset noticed by the reviewer is not too extreme, considering both age and elevation error bars.

Line 245 – evidence not evidences.

Changed accordingly

Line 326 – as not than.

Changed accordingly.

Fig 7 – what does Ma stand for? Why is the land black in this fig? It makes it harder to interpret than it should be!

Ma stands for Mangrove swamp. We changed the caption of the figure and included this missing information. The land is black in most modeling figures, so we are not changing it. In this figure, we indicate that land areas are filled in black color.

Line 446 – due to. I'm not sure you can compare a small island community extracting water to an Asian megacity here.

True, it is a comparison that gives an idea to the readers of the magnitude involved here, but caution is needed. So, we added this incipit to our sentence: "Notwithstanding the obvious differences in patterns and causes of subsidence".

Line 490 – most recent part not last part.

Changed accordingly.
Line 512 – delete important.

Changed accordingly.
Line 566 – use and instead of comma, and measurements instead of measurement.

Changed accordingly.

Line 592 – delete 'which concerns'.

Changed accordingly.

Lines 602-607 – I'm not sure about the section on correcting GIA predictions using different rates of VLM. Ok so the different models require different VLM corrections, but how do you know which is correct? There is no way of knowing. You need an independent measure of VLM. Is there anything else in the coastal geomorphology that might suggest the area is subsiding or uplifting long term? I don't think you can leave this section as it is without making some attempt to validate your conclusions (or you need to state more clearly that either positive or negative VLM is equally likely).

Unfortunately, other than the GPS stations cited (that are at odds with each other), we do not have any hard constraints on VLM. Many authors consider this area "stable", but we felt that this would be too simplistic in absence of clear indications. For this reason, we kept the paper open. We hope that, by restructuring the last part of our conclusions, our rationale is more clear.
* * *
Reviewer #3 (RV3)

Overall, authors do not clearly define the main impetus for this work nor do they offer specific insight into future research directions, implications of results for the Holocene sea-level history beyond the Spermonde Archipelago.

We thank RV3 for this comment and point out, that indicating future sea-level predictions or setting this new data in a broader context is not the aim of this study. The aim of the study is to show new data from the Spermonde Archipelago and compare this new dataset with previous studies from the same location and new GIA models, to unravel the existing inconsistencies in the RSL history between the three studies and to widen the knowledge of the RSL history in this rarely studied region.

It is not clear until the conclusion that the purpose of the GIA model-data comparison is not to identify glacio-eustatic contributions to late Holocene sea-level nor to investigate the evolution of late Holocene sea-level but to identify a best-fitting GIA model that could be used to refine predictions of current and future sea-level trends. A best-fitting model is ultimately not identified, nor are future directions. I therefore recommend major revisions to this manuscript to improve clarity.

We tried to streamline the last part of our conclusions also taking into account this comment.

The introduction does not establish why it is important to reconstruct sea-level changes during the Holocene, a time of transition between glacial and interglacial climates, nor does it emphasize why SE Sulawesi was selected for analysis, the power of microatolls as a proxy for past sea-level position on multiple timescales, and why evaluating data in the context of GIA model reconstructions of past sea-level is critical to evaluating global mean sea-level from local sea-level reconstructions in the past and the present.
We generally explain the importance of Holocene sea-level studies in the lines 55-63 and further, SE Sulawesi was not selected as study region. We further decided to explain the use of microatolls as sea-level index points and the use of GIA models in the methods. We decided, that technical things like these are better placed in the methods part, subdivided into their own topics, which isolates the introduction from the methods and gives the reader the chance to first indicate what this paper is about and then see how we conducted our study.

While the sea-level index points are combined with 3 previous studies from the vicinity of Makassar, there is no discussion of the results in the context of Late Holocene sea-level reconstructions from SE Asia and the South Pacific (e.g. Hallmann, 2018), or other Holocene SL reconstructions derived from microatolls.

We agree with RV3 that we did not discuss and compare our study and the studies from Mann et al., 2016, De Klerk, 1982 and Tjia et al., 1972 to other studies from the broader region of SE Asia and the South Pacific. It was our aim to compare only studies from the same region, to extend the RSL information in this location and to evaluate if the data from De Klerk and Tjia agrees or disagrees with our new data. It was not the aim to compare the new data from the Spermonde Archipelago to entire SE Asia and the southern Pacific region as this was already done by Mann et al., 2019 who indicated different data inconsistencies in several locations in SE Asia where the Spermonde Archipelago, (due to the data from De Klerk, Tjia et al and Mann et al., 2016) is one of these regions that needed more high-quality data (our study) to improve the knowledge of the local Holocene RSL history. A quote is mentioned in line 352 to 355.

How do RSL results compare with other reconstructions and what are the limitations of the previous works? As written, it does not seem to be anchored to a clear history of previous work (regional, global, Holocene) for readers to critically evaluate the importance and significance of the results that it presents, nor is it framed as novel or distinct form previous work in terms of its methods, study site, etc other than the introduction of new index points and extensive GIA modeling.

This analysis that is asked for in this comment, was already published by Mann et al., 2019 and is therefore not the aim of this study. In Mann et al., 2019 only three GIA models were compared to the different data sets, also to the 3 sets from the Spermonde Archipelago and we aim to extend this dataset with new data and a higher amount of GIA model outputs to improve the RSL history in this study area. It was a successful study as we can support the data by Mann et al., 2016 and discuss new reasons for the inaccuracy of the dataset from De Klerk, 1982 and Tjia et al., 1972. Further, we improved the previous studies in this location by a comparison to more GIA model outputs and can implicate that tectonic is not the reason for the difference in the RSL elevation results.

I would restructure the first 75% of the introduction as follows:
1. Statement on importance of reconstructing Holocene ice-sheet and sea-level response to an interglacial climate. Succinctly state why this is relevant to accurately and precisely predicting timing and rates of future ice-sheet and sea-level response to the present warming climate. Why are you presenting new Late Holocene sea-level data and GIA models?
2. Briefly, describe state of knowledge from far-field, Indo-Pacific Holocene sea-level reconstructions – any trends or outstanding questions. Why are far-field records important? How you reconstruct sea-level index points using microatolls and what are their advantages/disadvantages relative to other sea-level indicators?
3. How do you expect GIA to influence local sea-level histories in this region and why it is necessary to correct local sea-level histories for the influence of GIA and vertical land motion due to tectonism in ordered to evaluate glacio-eustatic sea-level changes? As you already mention in the introduction, determining rates of subsidence/uplift due to regional tectonics by accurately estimating past sea-level is a circular problem discussed at length in Creveling et al. (2015). *
4. Why was SE Sulawesi selected for analysis and what steps did you take (as written) to generate an accurate RSL reconstruction?

While we thank RV3 for this suggestion, we decided to make only minor changes to the introduction. About 1), we say briefly why this kind of study is important in the second paragraph of the introduction. A reviewer of the former version suggested us to downplay the "past for future" angle, so we will keep our rationale as it is now. About 2), most of the description the reviewer is asking is shifted to the first section of the methods. This was also done in response to a previous round of review. About 3), we also tried to insert some considerations on this point in the second paragraph of the introduction. About 4), we address this point in the first section of the "Regional Setting". It was shifted there after a previous comment from a reviewer, so we will not shift it to the introduction.

Definitions: In the intro, define terms such as relative (local) sea-level (RSL) (Line 49), and what you mean by "eustatic" sea-level (Line 45 vs 50).
We slightly restructured the first paragraph of the introduction to clarify what is eustatic and what is local sea level. We assume that readers of Climate of the Past will have a training in geoscience/climate science, so this brief reminder of concepts is enough, without entering detailed descriptions of eustatic and relative sea level concepts that are widespread in the literature.

On Line 45, you describe "globally averaged" sea-level, or GMSL (which is includes contributions from thermal expansion and changes in global ice-volume), but elsewhere (e.g. Line 50) you use "eustatic" to describe "glacio-eustatic" or "ice-equivalent" changes in sea-level in response to transfer of mass between ice and ocean (Mitrovica and Milne, 2002; Milne, 2015 *Handbook of Sea-level Research*, citations therein). It is my understanding that all three of the phenomena listed in Lines 51-54 to explain the common observation that far-field sea-level reconstructions record a mid-Holocene highstand fall under the definition of GIA (see concise explanation of equatorial syphoning/GIA trends and in Dutton et al, 2015 *Science* in addition to Kopp et al., 2015, Mitrovica and Milne 2002, Milne and Mitrovica 2008, etc.). GIA processes include deformational, gravitational and rotational effects driven by the transfer of mass between ice and ocean that can cause local RSL changes to depart significantly from the GMSL curve or the response of the solid Earth and gravity field to the climate-driven surface ice- water mass redistribution (Milne and Shennan, 2013). Syphoning, changes in gravity due to surface ice-water mass redistribution and solid Earth deformation are all driven by GIA
We modified our wording to make it more clear that GIA includes syphoning and rotational feedbacks. Thanks for pointing this out.

Line 44: I think that sections must be numbered. https://www.climate-of-the-past.net/for_authors/manuscript_preparation.html
We defer to the copy-editing process of Climate of the Past for this aspect.

Line 56: You repeat the definition of the RSL acronym again.
Thanks, we changed the text accordingly, and keep using the acronym.

Line 65: To reconstruct paleo RSL, we measured the age and elevation of microatolls, ie…Line 71: fossil ones, that we surveyed and dated using radiocarbon.
Accepted

Methods:
Lines 119 – 131: A conceptual figure or a reference to one may be useful here to visualize how microatolls are used as a proxy and linked to tidal datums, indicative range, etc. in this study.
In the section "coral microatolls" we make extensive reference to the most widely cited (and recent) literature on microatolls.

Line 124: Please be more specific about what you mean by "extended periods of time" perhaps relative to the growth rate of the coral?
We deleted the reference to "extended periods of time".

Line 130: Please be specific about what you mean by short-term sea-level fluctuations.
Decadal to centennial, fixed in the MS.

Line 140 – 141. This is an important point.
We thank RV3 for this comment.

Line 141: Clarify your definition of indicative meaning.
Following this comment, we decided to give a brief hint to what the indicative meaning is directly in this sentence. We then give a proper reference in the first lines of the "Paleo RSL calculation" section. There, we expanded the indicative meaning description with respect to the previous version.

Lines 117-141 General comment: Methods are clearly outlined. Assumptions made in using microatolls to reconstruct sea-level are not (e.g. as referenced in McLean et al., 1978). What assumptions go into assigning a reference water level to the coral's highest level of survival? Is the relationship between microatoll elevation and tidal cycle the same over time and across areas of the reef? Are all microatolls morphologically similar here and why is this a good field site?
The use of microatolls as good sea-level indicators is an accepted RSL measurement method by the sea-level community and explained or discussed in several previous publications, where some are cited in the previous section. We think that re-explaining the use of Microatolls for paleo RSL reconstructions would be redundant and out-of-scope for this MS. We further explain that the relationship between the microatoll elevation and the tidal cycle deviates due to site-specific characteristics, thus living microatolls should be used as modern counterparts to adjust fossil microatolls to the modern height of living coral and thus make sure living and fossil microatolls in the same site grew within similar conditions. We think this answers the question "Are all microatolls morphologically similar here?". With the HLC survey method, we exclude RSL elevation errors due to variabilities in the morphology of microatolls between the different study sites. The last question "why is this a good field site?" are indirectly answered in the "Regional setting" section, where we explain why this region is important to study.

Line 143: FMA's and LMA heights are the maximum (peak) height of the microatoll, correct? Or is it the average elevation surveyed across the top of the microatoll?
Yes, we always surveyed the highest rim of the microatoll. We clarified this in the MS.

Line 167: Replace reducing with relating.
We changed this word accordingly.

Line 177: How far away were these islands? Did you consider potential variations in the height of the geoid as per Woodroffe et al. (2012)?
Yes, we considered potential variations and checked the Geoid for differences, but published models do not show any appreciable variation.

Lines 204-222: Please explain further in the section on sampling and dating what kinds of samples you selected (slice of the microatoll? Hand samples?) and where you sampled from on the microatoll (the highest point on the microatoll or across it?).
We added a short explanation to this effect at the beginning of the sampling and dating section.

In general, how did you assign a radiocarbon age to a microatoll? Was there one date per microatoll or did you measure multiple dates to interpret an age (see distinction for U-series in Dutton et al., 2017)?
We obtained one age per microatoll, we clarified this point in the "Sampling and dating" section.

Additionally, please clarify what diagenetic screening you employed when analyzing coral preservation in advance of radiocarbon dating and report your XRD results (see more on XRD reporting in Vyverberg et al., 2018).

We dated corals with very high aragonitic content. We added a small section to the results to describe some samples affected by the presence of calcite, and we added the results of the XRD analysis to the Supplementary Material.

This information may also be useful in light of the documented erosion for most fossil microatolls in this study. Clarify your reference age for (a BP) – is the present defined as 1950 CE?
There is no need to clarify the BP convention, where 1950 is present.

Line 223: Please clarify here or in methods why you are predicting RSL with GIA modeling.
We believe that it is a very standard approach to predict RSL with GIA models and compare it with observed data. We added a short clarification at the beginning of the "Glacial Isostatic Adjustment" to avoid confusion.

Later, in the discussion, perhaps touch on the following: *Do you intend to convert RSL to GMSL via the extraction of the GIA signal at this location? Can any inferences of GMSL be made here? What steps would be needed to determine GMSL from your RSL results, and what are the challenges faced in converting RSL to GMSL via the extraction of the GIA signal at this location? Why don't you attempt to remove the GIA signal - provide clarification (e.g. further discussing implications of Fig. 11) as to why not. Elaborate on why evaluating GIA matters for Holocene/modern sea-level reconstructions and what the limitations to evaluating the GIA signal are here or in general.
The line of discussion was carefully selected in order not to overinterpred our data. Calculating GMSL from our data, with all the uncertainties embedded (VLM and GIA) would produce a spurious result, which would be of little interest.

Results:
Line 240: I see now that these are average radiocarbon ages as opposed to raw dates. I would clarify how many samples (dates) were analyzed to determine an age and how that data was evaluated for diagenetic alteration.
Our choice of words was odd. We dated 25 fossil microatolls and received one age per microatoll as explained earlier. We clarified also in this section.

Line 178, 248, 551: Please revise phrasing of "For which concerns" to "concerning …"
We rephrased these parts accordingly.

Line 240: Table 2: Following the equation on line 172, RSL estimates in Table 2 appear to be off by ~ 0.01 – 0.02 m. (ex – PS_FMA1 Suranti: RSL = -1.46 – (-0.74) + 0.2 = -0.52 m. In Table 2 it is reported as -0.53 m and using the numbers in Sheet 9 of SM1 RSL = -0.54 m. The excel sheet reads -0.53m using whatever rounding rules were applied in excel and the data correction of +0.014m. Furthermore, the Reference Water Level reported in SM1 is not always comparable to that reported in Table 2. For Suranti and Tambakulu it is as -0.72 m in SM1 but reported as - 0.74 in Table 2. Please address rounding and reporting discrepancies.
This was probably a glitch that remained from a previous version. Now SM and tables in the text coincide.

Table 3: Why is erosion error not included in Table 3, when it is included in SM1 Sheet 4 for FMA8 – 11 (Panambungan)? The erosion factor seems to be incorporated into RSL for those points in Table 3; without it the RSL values calculated in Table 3 are lower by 0.2m.
FMA 8-11 were published in Mann et al. The erosion error was already included in the original paper. As we took their elevation values for both LMA and FMA, we decided to keep them as they were provided. We wanted to visually separate our data table from the tables containing data from the other authors and therefore did not split the elevation calculation and used the elevation data as it was given by the original papers.

Discussion:

Line 318: Please clarify by how much HLC changes instead of "HLC changes substantially."
We changed the sentence to avoid the word "substantially"

Line 345: I would mention what "sea-level data" refers to. It is only mentioned in the caption of Figure 7 that the earlier works used different sea-level proxies.
We changed "sea level data" into "paleo sea-level observations". This should clarify what we mean.

Line 349: Be careful about the use of "significantly" here and throughout. Is the difference statistically significant?
No, it is not statistically significant but there is a big difference in the elevation results. To make this clearer we substituted "significantly" with "conspicuously" in this line.

Line 358: I would elaborate on the differences between the proxies used in these different studies. How does the precision vary between them? Looking at the uncertainty bars in figure 3, the De Klerk and Tjia sea-level index points tend to be higher than those reported in Mann and this study, but they are also less precise. Several points from this study/Mann fall within the bounds of vertical error for points from De Klerk and Tjia.
De Klerk published only one index point and Tjia et al. only report limiting data points. Figure 3 shows that these marine and terrestrial limiting indicators are, by definition, less precise than the index points presented by Mann et al and in this study. Thus, the difference is: these marine/ terrestrial limiting points presented by the two studies give only limiting indications on sea level, and cannot be equated to index points. In our criticism of these older datasets we address these points.

Line 368: What additional data would need to be explored to evaluate the tectonics hypothesis?
To clarify, we changed this sentence into: "While this is a possibility that would need further paleo RSL data to be explored (expanding the search of RSL indicators beyond the islands of the Spermonde Archipelago) [...]"

Line 373: Are the De Klerk coral data collected from coral in growth position? (in situ)
There is little information on these deposits, they were published long ago and reporting standards have changed since. Details for the coral at Tanah Keke do not allow to assess whether it is in situ or not. For the other corals and shells, they seem more to be described as 'accumulations' so not in situ.

Lines 519 – 560. It is to be expected that there is a range of highstand predictions that vary in space and time depending on GIA model ice and earth parameters, and it is clear that the fit between predicted and observed RSL also varies depending on the GIA model parameters as well as the assumed tectonic history. Is the main takeaway that the ICE5g model is not a good fit because, regardless of the tectonic history, the peak highstand predicted by ICE5g does not match the peak in the observed RSL data? What is the main outcome of this data-model comparison, or what steps can be taken to better compare them in future studies? The purpose of this comparison was not clearly defined in the first place, though the importance of identifying GIA models that best fit Late Holocene data to improving model predictions of current and future sea-level changes is explained on line 553.
We tried to streamline the last part of our conclusions also taking into account this comment.

How does the choice of GIA model affect the interpretation of RSL index points made in this study? What can the reader conclude about late Holocene sea-level form the data-model comparison described in these latter sections and the earlier comparison of data between this and previous RSL reconstructions? See previous comments on line 223 regarding inferences of GMSL at this location.
The choice of GIA models does not affect in any way the interpretation of RSL index points, and we believe that this is very clear in the paper.

The discussion of Late Holocene RSL and Fig. 12 seemed to end abruptly. Please elaborate on why the results in Fig. 12 are widely relevant to modern sea-level estimates.
We added an explanation at the end of the section, providing an example.

Line 572: Specify a gradient in elevation
This was extensively discussed above, we feel it would be redundant to repeat it here.

Figures:
General comment: All figures have simple and elegant layouts. Fonts are legible and colors are clear. Sections are clearly marked and figures are overall helpful and easy to follow.
We thank the RV3 for this comment.

Figure 1: in 1b, I would not combine red and green-colored dots to make the figure accessible to color-blind readers. Perhaps try a dark boarder to yellow dots in c – i to make them more legible.

Figure 3a: Have you tried making the symbols slightly different between datapoints from this study and from Mann? As mentioned earlier do not include red/green together.
We would rather not change the symbols, as they represent a standard for sea level studies. We changed the colors.

Figure 4a: I would identify the sites analyzed in this study with a (*) next to the name. Specify in the caption at least once that you mean individual microatolls instead of "individuals."
It is not clear what the reviewer means by "sites analysed in this study". We specified that we studied single microatolls in the caption.

Figure 5: Change Red/Green combination.
Done.

Figure 11: Change Red/Green combination. The four boxes in this figure are missing panel letters (a – d).
Done

Figure 12: I would mark the position of the Spermonde Archipelago on the map for reference.

Done
* * *
Reviewer #4 (RV4)

Authors critically re-evaluated reported index points by De Klerk (1982) and Tjia et al. (1972) and suggested to reconsider sediment interpretation as high-magnitude storm deposits and until further field investigation exclude them from sea-level compilations.
I also carefully reviewed authors' responses to comments by two anonymous reviewers and concluded that the manuscript was significantly improved since its original submission and that authors critically addressed reviewers concerns and suggestions.
We thank RV4 for this summary and comment.

I suggest that the manuscript will be considered for publication after few minor revisions.

1. In the Abstract authors state that they are reporting 24 new index sea-level points (line 38). However, in the Conclusion the authors report 25 index points (line 556). It is my understanding, that microatoll PB-FMA 4 index point was rejected. Please clarify.

This is true and we added the information that one index point was rejected to the sentence.

2. I suggest to add indexes "a" and "b" to the panels on Figure 8 to be consistent with other figures format.

We agree with RV4 and changed the figure and the capture accordingly.

3. I suggest to add indexes a, b, c, d to Figure 11. Text references to Figure 11 have already include the appropriate indexes (lines 530, 535, 539, and 542).

We thank the RV4 for this suggestion and changed the figure panels and the caption accordingly as we simply missed this.

In addition, I agree with R1's comment 2 regarding the anthropogenic subsidence on Barrang Lompo island being the major reason for a low rate of sea level rise. Since the instrumental data to support the proposed hypothesis does not exist, authors suggest that high rate of coastal erosion on the island could be indirect evidence of human impact and propose to further investigate this idea or leave the question open inviting other plausible explanation of the low rate that mismatch the regional sea level trend.

We take this comment as an approval of our choice to leave the discussion on this point open.

In the summary, I believe that the manuscript presents valuable data and paleo sea-level reconstruction using best-fit GIS model and is suitable for publication in CP. Analysis of ice models beyond the study area empathizing the need for GIA correction as essential for estimate of eustatic sea-level changes and future predictions presents an interest to a broader scientific community

Again, we thank RV4 for this comment.

**Late Holocene (0-6ka) sea-level changes in the Makassar Strait, Indonesia**

[revised manuscript text omitted]
 modelling results is shown in Figure 5Figure 5 and Figure 6Figure 6. While all models predict a RSL highstand in the Spermonde Archipelago (Figure 5Figure 5a), the RSL histories predicted by each model show significant differences. ICE5g, in fact, predicts the RSL highstand occurring ca. 2.5 ka later than ANICE and ICE6g. The maximum RSL predicted by ICE5g and ICE6g is higher than the one predicted by ANICE. ANICE is the only ice model for which some Earth model iterations do not predict a RSL highstand, but a quasi-monotonous sea level rise from 8 ka BP to present.

[Figure]

*Figure 5: Results of the 54 GIA model runs for the Spermonde Archipelago, a) last 9 ka. Dots indicate the points at which the*
*maps in Figure 6Figure 6 have been extracted. b) last 16 ka, representing the full time extent of the models. The eustatic sea*
*level for each ice melting scenario is available in SM2. The Jupyter notebook used to create this graph is available as SM2.*

[Figure]

*Figure 6: Relative sea level at 5 ka (left) and 7 ka (right) as predicted by three among the GIA models used in this study. See*
*Table 1Table 1 for the definition of the mantle viscosity here labelled as "Visco1".*

Discussion

The dataset presented in Table 2Table 2–4 and shown in Figure 3Figure 3a–c and Figure 4Figure 4 
[revised manuscript text omitted]

[Figure]

*Figure 11: Comparison between RSL observations and predictions from GIA models (see Table 1Table 1 for model details). Red,*
*green and blue lines represent, respectively, ANICE, ICE5g and ICE6g models. Black lines identify best fitting models. The*
*different panels (a-d) show different tectonic corrections applied to the observed RSL data. The Jupyter notebook used to*
*create this graph is available as SM2.*

Paleo to modern RSL changes

The different possiblebest matches between paleo RSL data and GIA models shown in Figure 11Figure
11 have a broader significance concerning rates and patterns of modern changes in relative sea level
at broad scale. In fact, GIA effects need to be taken into account in the analysis of both tide gauge and
satellite altimetry data (see Rovere et al., 2016 for a review). One way to choose the GIA model(s)
employed for this correction is to select those matching better with Late Holocene data.

To make an example of how different modelling choices propagate onto modern RSL estimates, in
Figure 12Figure 12a–c, we show the modern rates of GIA VLM predicted GIA predicted by three models
across Southern and Southeast Asia matching different assumptions on VLM (as shown in Figure 11).

by the three different models highlighted in Figure 11 as best matching with our data under different
vertical land-motion assumptions. The difference between the two most extreme models matching
with our data is within -0.3 and -0.5 mm/a (Figure 12Figure 12d), and it appears widely relevant also
within the broader geographic context included in our models.

For example, the values shown in Figure 12d show that ICE6g-VM6-60km predicts faster modern GIA
rates than ANICESELEN-VM1-60km for India and Sri Lanka. As these rates would need to be subtracted
from the data recorded by a tide gauge, this would have an effect on any attempt of decoupling the
magnitude of eustatic vs other land motions at that tide gauges in that area.

[revised manuscript text omitted]

In this study, we are not favor one model over the others nor claim that
our model ensemble is a complete representation of the possible variable space. We use the example
of the Spermonde Archipelago to highlight how
Holocene RSL data, coupled with GIA models, can inform  on two
aspects that are ultimately of interest for coastal populations. First, they may help defining local
subsidence rates beyond modern technologies. It appears that, for the Spermonde Archipelago, longterm subsidence, tectonic stability or slight uplift are all possible. To settle this uncertainty, instrumental measures and more precise Common Era sea level datasets should represent a focus of future sea-level research in this area. Second, we showed here that matching GIA model predictions with Late-Holocene RSL data is useful to constrain which models might be a better choice to predict ongoing regional rates of GIA. While we do not have a definite "best match" for the Spermonde Archipelago, we suggest that iterations of ICE6g and ANICESELEN fit better with our data, and might produce more reliable GIA predictions than ICE5g, that seems not to match as well as the other two.

As a final remark, we highlight that GIA needs to be accounted for to correct tide gauge data and derive current rates of eustatic sea-level changes. Under this perspective, disentangling which combination of Earth and ice models produces best-fitting RSL histories with late Holocene data is central in order to improve our understanding of future sea-level changes. In order 
[revised manuscript text omitted]

---

## Author Response (AR3)

Dear Editor,

We would like to thank you for your work on our MS. We feel that your work and that of the reviewers helped us greatly improve the original MS we submitted to *Climate of the Past Discussions*. Before answering your final (minor) queries, we highlight the following points to clarify the most important changes from our original submission.

1)  From the *Discussion* paper, we increased the number of GIA models that we use to compare predicted RSL histories to our observations. This leads to more robust considerations.
2)  We did a re-analysis of our original data, also in light of more precise tidal datums (we added one co-author who helped us on this aspect). As a result, the elevation values change slightly (few centimeters in most cases) from the original submission to the final version.
3)  We extended the number of analyses we perform on the combination of GIA and RSL data. To do this, and to increase transparency in our methods, we share a series of Jupyter Notebooks under a separate DOI. We kindly ask that bugs or suggestions for improvements are reported to arovere@marum.de.
4)  In general, we followed the constructive comments of the reviewers. This led to large modifications of the text and figures in the paper from the original MS published in the *Discussion* forum.

All our data, models and code used to analyze them are available and open-access. We welcome any feedback and suggestions for improvements readers might have.

In this latest version, we re-read carefully every paragraph trying to make it more clear for readers who might not be experts in sea-level studies. We corrected the English to the best of our possibilities.

We found one (minor) mistake in one formula in our excel files, that was affecting by 3 cm our measurements but not the paleo RSL calculations. We corrected the SM and tables in the paper accordingly. We re-checked all formulas and scripts, and now they are correct to the best of our knowledge.

In the following, the Editor's comments are highlighted in gray, and our response follows.

Rev#2 on Fig3: I agree with reviewer 2 that you should consider to plot eroded data differently. I found not that easy to see the difference with other data. Making the figure larger would also make the figure more readable.

Rev#3 on Fig3: There is obviously a problem of readability of this important figure. I understand that you prefer keeping the symbols, but there should be way to improve this figure. Making it larger as suggested above might be part of the solution but not only.

Finding a suitable graphical representation of Figure 3 is challenging, and we have changed the layout of this figure several times already. The problem is that every time we manage to represent one aspect particularly well, another one gets lots. We propose one last version. We hope this is enough to give a coherent overview of our data. Readers can always download our extensive supplementary and make their plots. Overall, we also added throughout the MS reminders that eroded microatolls should be regarded with more caution. Despite this, they still are sea-level index points, reconstructed with the highest possible scientific rigor, therefore we don't think it is appropriate to take them out of Figure 3 as one reviewer suggested.

Rev #3 comment on Lines 117-131: Please point to one or several specific conceptual figures in the references that you cite in the Methods section. For example: (Figure z in Doe et al 2001).

Rev#3 comment on Line 117-141: Related to the above comment, your reply state that it is not necessary to outline the assumptions made in using microattols to reconstruct sea-level change. I believe you can provide one or two sentences to explain these concepts as Climate of the Past is not a specialized journal in sea-level reconstruction, and your paper is definitely out of the zone of expertise of many readers. Another option is to add something as a Supplement.

In the text, we now point to two well-known references. Overall, we reworded and modified the first three paragraphs of section 4.1, trying to describe microatolls to a broader audience. We hope we managed to make this section more clear.

Rev#3 on Line 177: It is one thing to answer to this question in your reply, but I feel you should also add something in the text about this comment. It is quite simple to address.

This comment was referring to the study of Woodroffe et al., 2012, where microatolls were measured with GPS with respect to the ellipsoid, and had to be referred to MSL through a geoid model. Differences in the geoid would then cause a difference in the calculated elevation with respect to MSL. Our approach refers directly to MSL, so we address this simply without adding a long explanation of the different survey techniques and their implications. We believe that this is out of the scope of this MS and would unnecessarily confuse the reader. This is why we refrain from putting a further explanation of this in the text, we hope the Editor will agree with our decision.

Rev#3 on inferring GMSL: why don't you write somewhere that your data are not suitable for calculating GMSL?

We now state this clearly at the beginning of section 6.6

Rev#3 on Table 3: Why don't you add a note in the table caption that the errors are included in the Mann et al paper?

Done

Rev#3 on line 572: I also think you should specify "gradient in elevation" here, because even if you extensively discussed this above, it is not yet mentioned in the conclusion. Many readers only read parts of papers, and conclusion is one of the most popular parts. Your conclusion will gain in clarity and it is only 2 words to add.

Done

Rev#3 about Fig4: I guess your modification adds more confusion. I suggest writing: "Box plot of the HLC elevation measured in the Spermonde Archipelago; "n" indicates the number of living microattols that were surveyed on each island.". You can easily add a (*) next to Tambakulu and Sanrobengi that you measured yourself (as opposite to Mann)

We revised Figure 4, adding also the microatoll thickness reported by Mann et al. This gave us the possibility to discuss a bit more in detail a few things in the MS, and to show a bit better our data.

As for the suggestion of Reviewer #3 to restructuring the introduction, I understand that you changed an earlier version of the introduction according to previous reviewers recommendation and that it is difficult to reconcile reviewer's #3 suggestion with earlier recommendation.

Thank you

[revised manuscript text omitted]
 that microatolls grow upwards until they ir polyps reach MLWthe lower part of the
tidal range., and sAt this pointuccessively, they keep growing horizontally at the same elevation
forming "atoll-like" structures (Figure 1 in Scoffin and Stoddart, 1978 and Figure 8.1 in Meltzner and
Woodroffe, 2015) that can widen up to several meters (Figure 1 in Scoffin and Stoddart, 1978 and
Figure 8.1 in Meltzner and Woodroffe, 2015).

In the most standard definition, microatolls live at Mean Lower Low Water (MLLW), but their living
range can span from Mean Low Water (MLW) down to the Lowest Astronomical Tide (LAT) (Mann et
al., 2019). In general, this restricted range of formation reflects the fact that microatolls grow upwards
until their polyps reach MLW, and successively keep growing horizontally at the same elevation (Figure
1 in Scoffin and Stoddart, 1978 and Figure 8.1 in Meltzner and Woodroffe, 2015). If sea level falls below
LAT, the coral polyps diedesiccate and die, retaining their carbonate calcium fossil skeleton and their
morphology only (Meltzner and Woodroffe, 2015). Due to thisSince they can survive within a narrow
range related to tidal datums, characteristic, fossil microatolls are often considered as an excellent RSL
indicator (when found in good preservation state) as they constrain paleo RSL within a narrow range
(Meltzner and Woodroffe, 2015).

The methodology to measure paleo RSL is based on the microatoll characteristic to always live within
the range of MLLW to MLW. This behavior does not change over time, thus modern microatolls live in
similar tidal datums and provide the same relationship to MSL as fossil microatolls, within an
uncertainty range. Based on this, the elevation of the fossil microatoll can be referenced to modern
MSL and by subtracting the elevation of modern microatolls with respect to MSL that is called the
height of living coral (HLC), RSL can be derived. Fossil microatolls can also be assigned an age, either
by ¹⁴C (Woodroffe et al., 2012) or U-series dating (Azmy et al., 2010). Recent studies showed that the
accurate measurement, dating and standardized interpretation of coral microatolls has the further
potential to detail patterns and cyclicities related to short term (e.g. decadal to centennial) sea level
fluctuations (Meltzner et al., 2017; Smithers and Woodroffe, 2001; Kench et al., 2019).

While the relationship of coral microatolls with the tidal datums described above is often maintained,
several authors (e.g. Mann et al., 2016; Smithers and Woodroffe, 2001; Woodroffe et al., 2012) pointed
out that deviations from microatoll living range and tidal datums may occur due to site-dependent
characteristics, such as wave intensity, tidal ranges and broader reef morphology (Meltzner and
Woodroffe, 2015). It is also worth highlighting that a tide gauge with long enough time series might
not be available at remote locations where microatolls are often found. Therefore, it is both more
practical and more accurate to reconstruct paleo RSL at the time of microatoll life starting from the
height of living coral microatolls (Height of Living Coral microatolls, HLC). Under the assumption that
tide, wave, and reef morphology did not change significantly in time, Thisthis allows determining the
paleo RSL associated to fossil microatolls that were living onin the same geographical setting as
modern ones (i.e., the same island or group of islands). For this reason, in this study, we sampled both
fossil and living microatolls elevations, and we determined the indicative meaning (i.e., the correlation
with sea level) of the fossil microatolls from the HLC rather than to tidal datums.

As fossil microatolls are composed of calcium carbonate, they can be assigned an age, either with ¹⁴C
(Woodroffe et al., 2012) or U-series dating  (e.g., Azmy et al., 2010). Recent studies showed that the
accurate measurement, dating and standardized interpretation of coral microatolls have the further
potential to detail patterns and cyclicities related to short-term (e.g. decadal to centennial) sea-level
fluctuations (Meltzner et al., 2017; Smithers and Woodroffe, 2001; Kench et al., 2019).

## 4.2   Elevation measurements

Fossil and living microatoll (respectively, FMA and LMA) heights were surveyed on Sanrobengi,
Kodingareng Keke, Bone Batang, Suranti and Tambakulu (Figure 1Figure 1c–i) with an automatic level.
FMA and LMA heights were always taken on the top microatoll surface. Elevations were initially
referenced to locally deployed water level sensors (Seametrics PT2X) acting as temporary benchmarks.
Locations of water level loggers are shown as stars in Figure 1Figure 1c–il (stars), and logged water
levels are reported in SM1. These sensors were fixed to either jetties or living corals close to the survey
sites and logged the tide levels at 30-second intervals. To exclude differences in the Geoid over the

[revised manuscript text omitted]